# CoEvol-NO: State and Coordinate Co-Evolution with an Error-Driven Predictor-Corrector Paradigm for Neural Operator Transformer

**Jianqiao Zeng** [1 2]  **Ruocheng Wang** [3]  **Yanzhi Liu** [3]  **Hao Xiong** [1 2]  **Junchi Yan** [3]

## Abstract

The applicability of neural operators remains limited by the challenges of handling complex physical conditions in real-world settings. We argue that this limitation stems from the inherent trade-off between geometric sensitivity to complex boundaries and the dynamic memory required for long-term physical evolution. Instead of directly addressing these ad-hoc constraints, we propose that modeling the co-evolution of latent states and mesh sequences offers a more fundamental solution: latent states accumulate physical features across layers, while mesh sequences provide real-time geometric feedback. To this end, we propose CoEvol-NO, a co-evolutionary framework where the latent state and mesh sequence are updated jointly . To enhance the capacity of latent state evolution, we introduce the classical Predictor-Corrector (PC) paradigm, formulating the layer-wise evolution as a meta-learning process with an implicit objective optimized during the forward pass: the Predictor generates a tentative target, while the error-driven Corrector refines the persistent state via gradient-based optimization. Furthermore, our theoretical analysis reveals that the widely used *direct substitution* and *residual update* paradigms are essentially first-order approximations of this error-driven correction under different loss assumptions. We also prove that CoEvol-NO achieves strict linear time complexity. Extensive experiments across five standard benchmarks and two large-scale industrial tasks demonstrate that CoEvol-NO achieves state-of-the-art (SOTA) performance.

---

[1]AI[3] Institute, Fudan University [2]Shanghai Academy of AI for Science [3]School of AI, Shanghai Jiao Tong University. Correspondence to: Hao Xiong <haoxiong@fudan.edu.cn>, Junchi Yan <yanjunchi@sjtu.edu.cn>.

*Proceedings of the 43rd International Conference on Machine Learning*, Seoul, South Korea. PMLR 306, 2026. Copyright 2026 by the author(s).

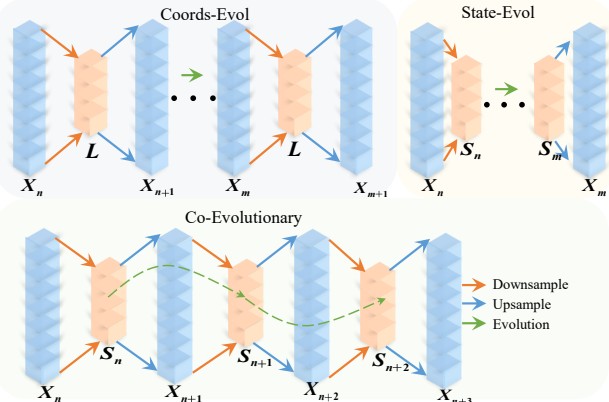

*Figure 1.* Comparison of different latent-space interaction paradigms: (a) Coords-Evol (Wu et al., 2024), (b) State-Evol (Wang & Wang, 2024), and (c) Co-Evolutionary.

## 1. Introduction

Solving Partial Differential Equations (PDEs) (Evans, 2010; Courant et al., 1967) constitutes the foundation of scientific discovery and engineering design, spanning domains from fluid dynamics to structural mechanics (Lynch, 2008; Gupta & Brandstetter, 2022). While Neural Operators have emerged as a powerful data-driven paradigm for such tasks (Pathak et al., 2022; Feng et al., 2025a; Cao, 2021),the application of standard Transformers (Vaswani et al., 2017) is often hindered by a critical bottleneck: the quadratic complexity $\mathcal{O}(N^2)$ of the self-attention mechanism (Kitaev et al., 2020; Xiong et al., 2021; Zhang et al.). In real-world simulations involving high-resolution meshes with hundreds of thousands of grid points, this computational cost becomes prohibitively expensive.

To circumvent this scaling limitation, a prevailing strategy involves compressing high-dimensional mesh data ($\mathbf{X}$) into a compact latent state ($\mathbf{S}$) (Prud'homme et al., 2002), which serves as a low-rank information bottleneck to achieve linear complexity (Lee et al., 2019; Wang et al., 2020; Jaegle et al., 2021). However, it is observed that existing methods often fail to fully leverage the potential of this latent representation due to suboptimal interaction mechanisms. For instance, Transolver (Wu et al., 2024) treats the latent state as a *transient* variable, recomputing it from the mesh at every layer (Coords-Evol). While effective for static geometry,

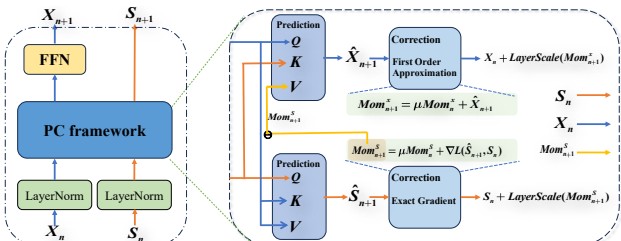

*Figure 2.* Schematic of the CoEvol-NO architecture. The model maintains a persistent latent state $S$ and a coordinate sequence $X$, which co-evolve through a Predictor-Corrector framework. The latent state is updated using a momentum-enhanced exact gradient of the correction loss, while the mesh sequence follows an efficient first-order approximation.

this design fails to maintain a persistent memory of physical evolution, limiting its performance on dynamic tasks like Navier-Stokes. Conversely, Latent Neural Operator (LNO) (Wang & Wang, 2024) encodes the input only once and subsequently evolves only the latent state (decoupled), effectively severing the connection with mesh details during deep processing. This leads to significant information loss in tasks requiring fine-grained spatial constraints, such as Elasticity.

It is posited that capturing the intricate spatiotemporal dynamics of PDEs requires a *co-evolutionary framework*, where the latent state **S** and the mesh sequence **X** are updated jointly. In this scheme, **S** accumulates physical dynamics across layers, while **X** preserves fine-grained geometric details. Drawing inspiration from classical numerical analysis, the *Predictor-Corrector (PC)* framework is adopted as the core design principle. Fundamentally, PC methods solve differential equations by alternating between two stages: an *explicit prediction* step that proposes a tentative solution, and a *correction* step that refines this solution by minimizing approximation error.

In this work, we propose **CoEvol-NO**, a novel architecture grounded in a Predictor-Corrector mechanism with error-driven updates (Kramer et al., 2024; Press et al., 1992; Feng et al., 2024). In the CoEvol-NO architecture, the "Predictor" step utilizes cross-attention to aggregate information to generate a tentative target estimation. Subsequently, the "Corrector" step refines the state based on the discrepancy between the current and predicted states (). This correction process is formalized as a *single-step gradient descent*, adjusting the state to minimize the residual error. Specifically, it is revealed that standard update rules in deep learning—such as direct substitution and residual connections (He et al., 2015)—are essentially first-order gradient approximations of the proposed optimization process under squared loss and dot-product loss assumptions, respectively. By employing the exact optimization update, CoEvol-NO achieves superior precision and stability. **Our contributions are:**

1) We propose CoEvol-NO, for the first time to our knowledge, establishing a co-evolutionary paradigm that updates both latent states and mesh i.e. point coordinate sequences jointly. Through an analysis of three architectural paradigms, we demonstrate that our approach addresses the limitations of the "state-evolve" paradigm (which struggles to fully utilize positional information) and the "Coords-Evolve" paradigm (which lacks dynamic memory).

2) We design a Predictor-Corrector (PC) block as the core update mechanism that employs a "Predictor" for target generation and an error-driven "Corrector" for state refinement. Theoretical analysis reveals that common residual and direct updates are first-order approximations of our exact gradient-based correction.

3) We theoretically prove that CoEvol-NO achieves linear time complexity. Experiments across five benchmarks (covering point clouds, structured meshes, and regular grids) and two large-scale industrial design tasks show that it achieves state-of-the-art performance.

**Remark.** It is worth noting that as verified in our experiments in Table 6, CoEvol-NO still preserve the ability to generalize to grids with different resolutions.

## 2. Related Work

### 2.1. Neural Operators

They learn mappings between infinite-dimensional function spaces. Pioneering works like DeepONet (Lu et al., 2019) and Fourier Neural Operator (FNO) (Li et al., 2021) and their variants (Wang et al., 2025b; Feng et al., 2025b; Guibas et al., 2021), have demonstrated remarkable capability in modeling PDE families. To handle complex and irregular geometries, Graph Neural Operators (GNO) (Li et al., 2023a) and related message-passing frameworks (Li et al., 2020a) treat the mesh nodes as a graph, parameterizing the integral kernel via local neighborhood aggregations. To further enhance the versatility of such frameworks, GNOT (Hao et al., 2023) leverages a Transformer-based backbone to achieve a general-purpose operator capable of generalizing across varied geometries and mesh densities. Furthermore, ONO (Xiao et al., 2023) introduces orthogonal attention mechanisms to improve the stability and representation capacity of the learned solution operators.

### 2.2. Taxonomy of Efficient Architectures

To mitigate the quadratic cost of self-attention, diverse linear-complexity mechanisms have been proposed. One prevailing strategy imposes a Low-Rank Bottleneck to compress high-dimensional mesh data for efficient global modeling. Within the scope of these low-rank architectures, existing methods can be categorized into two distinct paradigms.

*Table 1.* **Comparison of Architectural Paradigms.** We categorize existing linear operators into Coords-Evol and State-Evol paradigms. Our Co-Evolution paradigm resolves their inherent trade-off between geometric sensitivity and dynamic memory.

| Paradigm | Representative Method | Latent State Nature | Interaction Mechanism | Geometry (Mesh Info.) | Dynamics (Evolution) |
|---|---|---|---|---|---|
| **Coords-Evol** | Transolver (Wu et al., 2024) | Transient (Reset) | Re-encoding ($\mathbf{X} \rightarrow \mathbf{S}$) | **High** | **Low** |
| **State-Evol** | LNO (Wang & Wang, 2024) | Persistent (Isolated) | None ($\mathbf{S} \rightarrow \mathbf{S}$) | **Low** | **High** |
| **Co-Evolution** | **CoEvol-NO** | **Persistent (Evolving)** | **Joint $(\mathbf{S}, \mathbf{X})$** | **High** | **High** |

**Coordinate-Evolution Models.** This paradigm approximates attention directly on the discretized spatial mesh to maintain geometric fidelity. In general vision tasks, Set Transformer (Lee et al., 2019) and Nyströmformer (Xiong et al., 2021) utilize inducing points or landmarks to construct low-rank approximations. In the operator learning domain, Transolver (Wu et al., 2024) adopts this paradigm by dynamically slicing the mesh into geometric partitions. While these methods excel at preserving fine-grained positional information, they typically reset their features from the mesh at every layer. This lack of persistent memory limits their ability to accumulate the long-term physical inertia required for complex dynamics.

**State-Evolution Models.** Conversely, this paradigm compresses inputs into a compact latent bottleneck, decoupling the evolution from the spatial grid. General architectures like Perceiver (Jaegle et al., 2021) and TokenLearner (Ryoo et al., 2021) project data onto latent tokens for efficient processing. Latent Diffusion Models (Rombach et al., 2022) similarly rely on VAE-based compression. In operator learning, LNO (Wang & Wang, 2024) exemplifies this approach by evolving latent tokens in isolation. Although efficient for global dynamics, the disconnection from the mesh boundaries during deep evolution often leads to significant geometric information loss.

**CoEvol-NO.** Unlike single-carrier paradigms, our approach establishes a *Co-Evolutionary* framework. By synchronizing the latent state and mesh sequence via a joint Predictor-Corrector mechanism, CoEvol-NO addresses the dynamic memory of Coordinate-Evolution and the geometric information loss of State-Evolution.

### 2.3. Fast Weight Programmers and Meta-Learning

Recent research redefines the forward pass of neural networks as an online training process (Sun et al., 2024; Schlag et al., 2021; Zhang et al., 2025), a perspective rooted in the Fast Weight Programmer (FWP) paradigm (Schmidhuber, 1992). In this framework, a "slow" network with meta-learned weights programs "fast weights" (context memory) to minimize self-supervised objectives during inference, typically using simple linear operations such as outer products (Horn et al., 2017; Schlag et al., 2021; Yang et al., 2024). While methods like TTT (Sun et al., 2024) perform weight

updates per token along the sequence axis, CoEvol-NO introduces this perspective to neural operators through a layer-wise update strategy. This design effectively mitigates the computational bottlenecks of TTT regarding sequence length and allows for the integration of more sophisticated Predictor-Corrector (PC) blocks to achieve precise refinement of physical fields beyond simple linear mappings.

## 3. Preliminary

### 3.1. Problem Setup: Neural Operator Learning

We consider the problem of learning a non-linear operator $\mathcal{G}$ that maps between infinite-dimensional function spaces as widely studied in literature (Lu et al., 2019). Let $\mathcal{D} \subset \mathbb{R}^d$ be a continuous physical domain. The operator $\mathcal{G}$ maps an input function $a \in \mathcal{A}(\mathcal{D})$ to a solution function $u \in \mathcal{U}(\mathcal{D})$:

$$\mathcal{G} : a \mapsto u, \quad \text{subject to } \mathcal{F}(a, u; \xi) = 0, \qquad (1)$$

where $\mathcal{F}$ denotes the governing Partial Differential Equation (PDE) and $\xi$ represents the physical parameters. In practical scenarios, the functions are often observed on a discretized mesh $\mathbf{X} = \{x_i\}_{i=1}^{N} \subset \mathcal{D}$. The input is represented as a collection of point-value pairs $\{(x_i, a(x_i))\}_{i=1}^{N}$, and the goal is to find an approximate operator $\mathcal{G}_\theta$ that minimizes the risk $\mathbb{E}_{a \sim \mu}[\|\mathcal{G}(a) - \mathcal{G}_\theta(a)\|]$, where $\mu$ is a probability measure on $\mathcal{A}$ (Li et al., 2021).

### 3.2. The Predictor-Corrector Framework

The Predictor-Corrector (PC) framework (Ascher & Petzold, 1998; Press et al., 1992) is a classical iterative strategy in numerical analysis, e.g. for solving Ordinary Differential Equations (ODEs) in the form $\frac{dy}{dt} = f(t, y)$. To circumvent the high computational cost of implicit methods (e.g., Implicit Euler) while maintaining stability, PC methods decouple the integration into two sequential phases:

**Prediction**: An explicit step (e.g., Adams-Bashforth) generates a tentative estimate $\hat{y}_{n+1}$ based on previous states:

$$\hat{y}_{n+1} = y_n + \int_{t_n}^{t_{n+1}} \tilde{f}(t, y) dt, \qquad (2)$$

where $\tilde{f}$ is an extrapolating polynomial.

**Correction**: An implicit step (e.g., Adams-Moulton) refines the estimate by solving for the fixed point or minimizing the

residual error:

$$y_{n+1} = y_n + \frac{h}{2} \left( f(t_n, y_n) + f(t_{n+1}, \hat{y}_{n+1}) \right). \quad (3)$$

*Remark* 3.1 (**Error-Driven Correction**). The layer-wise evolution is defined by a trajectory $\mathbf{X}_0 \to \mathbf{X}_1 \to \cdots \to \mathbf{X}^*$, where $\mathbf{X}^*$ is the ideal representation. Each layer minimizes the local discrepancy $\mathcal{L}(\mathbf{X}_n, \mathbf{X}^*)$ via gradient descent:

$$\mathbf{X}_{n+1} = \mathbf{X}_n - \eta \nabla_{\mathbf{X}_n} \mathcal{L}(\mathbf{X}_n, \mathbf{X}^*). \quad (4)$$

In implementation, the inaccessible target $\mathbf{X}^*$ is substituted by a predicted estimate $\hat{\mathbf{X}}_{n+1}$. Consequently, the state update direction is determined by the gradient of the local prediction error.

## 4. Methodology

### 4.1. Overall Architecture: CoEvol-NO

To model the infinite-dimensional operator $\mathcal{G}$ efficiently, CoEvol-NO is introduced as a linear-complexity architecture grounded in the principle of error-driven correction. To bypass the $\mathcal{O}(N^2)$ bottleneck of standard self-attention, a latent state $\mathbf{S} \in \mathbb{R}^{M \times C}$ and a mesh representation $\mathbf{X} \in \mathbb{R}^{N \times C}$ ($M \ll N$) are maintained throughout the network. Diverging from existing methods that either recompute the state from the mesh or evolve the latent tokens in isolation, CoEvol-NO employs a co-evolutionary mechanism where $\mathbf{S}$ and $\mathbf{X}$ are updated jointly at each layer. This design ensures that $\mathbf{S}$ accumulates persistent physical dynamics while $\mathbf{X}$ provides fine-grained geometric feedback to guide the state refinement.

Unlike traditional approaches that treat latent representations as static features or transient variables, CoEvol-NO conceptualizes the layer-wise evolution as a co-evolutionary dynamical system. In each layer $t$, the current state $\mathbf{S}_{t-1}$ is refined via an error-correction process against a tentative target $\hat{\mathbf{S}}_t$ generated by the predictor. This error-driven refinement is tightly coupled with the update of the mesh sequence $\mathbf{X}$. The overall architecture is illustrated in Figure 2.

### 4.2. Update of Latent State S: Predictor-Corrector Co-Evolution

The core of CoEvol-NO lies in its state update mechanism. Lacking intermediate ground truth, we model the evolution of $\mathbf{S}$ as a self-supervised Predictor-Corrector (PC) framework. Formally, we conceptualize the sequence of states $\{\mathbf{S}_t\}_{t=0}^{L}$ as a discrete-time dynamical system that seeks to approximate an optimal physical representation $\mathbf{S}^*$. This representation is defined as a fixed point that ideally satisfies the underlying physical constraints of the operator. Under this framework, each layer $t$ aims to satisfy the following *refinement condition* (Informal) $d(\mathbf{S}_t, \mathbf{S}^*) \leq d(\mathbf{S}_{t-1}, \mathbf{S}^*)$,

where $d(\cdot, \cdot)$ denotes a distance metric on the latent manifold. To achieve this progressive refinement, the update process is decomposed into two distinct, alternating phases:

**Prediction Step: Generating the Approximation Target**
In this phase, a tentative target $\hat{\mathbf{S}}_t$ is generated to serve as a local proxy for the next state. A state predictor $f_t$ is defined to aggregate relevant physical information from the mesh representation $\mathbf{X}_{t-1}$ into the latent space:

$$\hat{\mathbf{S}}_t = f_t(\mathbf{S}_{t-1}; \mathbf{X}_{t-1}) \quad (5)$$

We adopt the Direct Mapping formulation for $f_t$, where the target is defined as the direct output of the predictor. This choice is motivated by the role of $S$ as a physical information bottleneck: an optimal representation should satisfy the consistency condition $S^* \approx f(S^*)$, representing a fixed point of the information extraction process. Justification and comparison with other targets and regularization properties are in Appendix B. We instantiate $f_t$ as a Cross-Attention module, where the state queries the mesh:

$$\hat{\mathbf{S}}_t = \text{Attention}(Q = \mathbf{S}_{t-1}, K = \mathbf{X}_{t-1}, V = \mathbf{X}_{t-1}) \quad (6)$$

Here, $\hat{\mathbf{S}}_t$ serves as the "next-step estimate," derived from the interaction between the global state and mesh details.

**Correction Step: Error-Driven Correction** In the correction phase, we refine the state from $\mathbf{S}_{t-1}$ to $\mathbf{S}_t$ based on the discrepancy between the current value and the predicted target. We formalize this update as a single-step gradient descent optimization.

We define a Correction Loss $\mathcal{L}_{corr}$ to quantify the deviation. A canonical choice, motivated by the Delta Rule, is the squared Frobenius norm:

$$\mathcal{L}_{corr}(\mathbf{S}_{t-1}; \hat{\mathbf{S}}_t) = \frac{1}{2} \|\mathbf{S}_{t-1} - f_t(\mathbf{S}_{t-1}; \mathbf{X}_{t-1})\|_F^2 \quad (7)$$

Drawing inspiration from error drivern architecture (e.g., TTT (Sun et al., 2024), Nested Learning (Behrouz et al., 2025)), we define the state update direction as the negative gradient of this internal objective. The updated state is:

$$\mathbf{S}_t = \mathbf{S}_{t-1} - \eta_t \nabla_{\mathbf{S}_{t-1}} \mathcal{L}_{corr}(\mathbf{S}_{t-1}; \hat{\mathbf{S}}_t) \quad (8)$$

Critically, since the target $\hat{\mathbf{S}}_t$ itself depends on $\mathbf{S}_{t-1}$ (via the Query projection), the gradient computation involves the Jacobian of the predictor. Using the chain rule, the exact gradient form is:

$$\nabla_{\mathbf{S}_{t-1}} \mathcal{L}_{corr} = (\mathbf{S}_{t-1} - \hat{\mathbf{S}}_t) - (\nabla_{\mathbf{S}_{t-1}} f_t)^\top (\mathbf{S}_{t-1} - \hat{\mathbf{S}}_t) \quad (9)$$

This formulation allows to capture complex, higher-order dependencies during the state transition. We discuss alternative objectives, e.g. the Dot-Product Loss in Appendix A.2.

### 4.3. Update of Sequence X: Gradient Approximation

Parallel to the state evolution, refining the mesh representation $\mathbf{X}$ is crucial for preserving local details. Ideally, $\mathbf{X}$ would follow the same rigorous optimization framework as $\mathbf{S}$. However, since $N \gg M$, computing the exact Jacobian for $\mathbf{X}$ is computationally prohibitive. To circumvent this, we propose a strategy based on Gradient Approximation, a detailed analysis of which is provided in Appendix A.2.2

Let us revisit the general gradient descent rule for a variable $\mathbf{Z}$ minimizing loss $\mathcal{L}$: $\mathbf{Z}_t = \mathbf{Z}_{t-1} - \eta \nabla \mathcal{L}$. By approximating the exact gradient with its primary first-order term (ignoring high-order curvature), we can recover distinct architectural paradigms depending on the objective:

**Case 1: Squared Error & Direct Substitution.** Assuming $\mathcal{L} \approx \frac{1}{2} \|\mathbf{X} - f\|^2$, the first-order gradient is $\nabla \mathcal{L} \approx \mathbf{X} - f$. With step size $\eta = 1$, the update becomes $\mathbf{X}_t = f$.

**Case 2: Dot-Product Maximization & Residual Learning.** Assuming $\mathcal{L} \approx -\mathrm{Tr}(\mathbf{X}^\top f)$ (maximizing alignment), the gradient is $\nabla \mathcal{L} \approx -f$. With $\eta = 1$, the update yields $\mathbf{X}_t = \mathbf{X}_{t-1} + f$, recovering the Residual Update paradigm fundamental to ResNets and Transformers.

This derivation unifies standard blocks as different first-order approximations of an underlying optimization process. In CoEvol-NO, we adopt the Residual Update paradigm due to its superior gradient flow properties.

Based on this insight, the update mechanism for $\mathbf{X}$ is designed to propagate refined global information back to the local mesh. A second Cross-Attention module is employed where the mesh queries the updated latent state. Crucially, the *Value* is set to the state increment $\Delta \mathbf{S}_t = \mathbf{S}_t - \mathbf{S}_{t-1}$, explicitly distributing the correction signal back to the mesh:

$$\mathbf{I}_t = \mathrm{CrossAtten}(Q = \mathbf{X}_{t-1}, K = \mathbf{S}_t, V = \Delta \mathbf{S}_t). \quad (10)$$

Finally, adhering to the residual paradigm derived above:

$$\mathbf{X}_t = \mathbf{X}_{t-1} + \mathbf{I}_t + \mathrm{FFN}(\mathbf{X}_{t-1} + \mathbf{I}_t), \quad (11)$$

where $\mathbf{I}_t$ represents the local geometric refinement driven by the global latent correction.

### 4.4. Optimizer Enhancements for Evolution

To mitigate slow convergence and gradient noise inherent in deep unrolled optimization, a *layer-wise momentum* mechanism is incorporated. Specifically, a velocity buffer $\mathbf{V}_t \in \mathbb{R}^{M \times C}$ is maintained to accumulate exponential moving averages of gradients across layers:

$$\mathbf{V}_t = \beta \mathbf{V}_{t-1} + \nabla_{\mathbf{S}_{t-1}} \mathcal{L}_{\mathrm{corr}}, \quad (12)$$

Finally, the latent state is updated using the accumulated velocity. We employ LayerScale (Touvron et al., 2021) to

implement learnable step sizes for each feature dimension:

$$\mathbf{S}_t = \mathbf{S}_{t-1} - \mathrm{LayerScale}(\mathbf{V}_t). \quad (13)$$

where $\beta \in [0, 1)$ is the decay factor. It acts as a *low-pass filter* that smooths the state trajectory $\mathbf{S}_0 \to \cdots \to \mathbf{S}_L$, effectively suppressing high-frequency oscillations and accelerating convergence along consistent descent paths.

### 4.5. Linear Complexity Analysis

To process high-resolution meshes where $N$ is large, maintaining linear complexity $\mathcal{O}(N)$ is imperative. We prove this by deriving the analytical gradient of the correction loss $\mathcal{L}_{\mathrm{corr}}$ with respect to the latent state $\mathbf{S}$ (derivation provided in Appendix A.1). Let $d_k$ be the scaling factor and $\mathbf{P} \in \mathbb{R}^{M \times N}$ be the cross-attention weight matrix. The exact closed-form gradient is:

$$\nabla_{\mathbf{S}} \mathcal{L} = (\mathbf{S} - \hat{\mathbf{S}}_t) - \frac{1}{\sqrt{d_k}} \Big[ \big( \mathbf{\Delta} - \mathrm{rs}(\mathbf{\Delta} \circ \mathbf{P}) \mathbf{1}^\top \big) \circ \mathbf{P} \Big] \mathbf{K}_X (\mathbf{W}^Q)^\top, \quad (14)$$

where $\mathbf{\Delta} = -(\mathbf{S} - \mathbf{P} \mathbf{V}_X) \mathbf{V}_X^\top \in \mathbb{R}^{M \times N}$ represents the projected error term, $\mathrm{rs}(\cdot)$ denotes the row-sum operator, and $\mathbf{1} \in \mathbb{R}^N$ is an all-ones vector.

**Complexity analysis.** Throughout the computation of Eq. (14), the complexity is dominated by three operations: (i) calculating $\mathbf{\Delta}$ via $(M \times C) \cdot (C \times N)$ multiplication, costing $\mathcal{O}(MNC)$; (ii) computing the Hadamard product ($\circ$) and row-sum on $M \times N$ matrices, costing $\mathcal{O}(MN)$; (iii) projecting via $\mathbf{K}_X$ through $(M \times N) \cdot (N \times C)$ multiplication, costing $\mathcal{O}(MNC)$. Crucially, as $M \ll N$, all operations remain strictly linear with respect to the mesh size $N$. By avoiding the instantiation of $N \times N$ attention matrices, CoEvol-NO scales efficiently to industrial-scale simulations with millions of mesh coordinate points.

## 5. Experiment

We conduct extensive experiments to evaluate CoEvol-NO on five well-established benchmarks (Elasticity, Airfoil, Pipe, Darcy, Navier-Stokes) and two challenging industrial-level design tasks (Shape-Net Car, AirfRANS), covering various geometries from 2D regular grids to 3D unstructured meshes. We benchmark against state-of-the-art neural operators, including FNO (Li et al., 2020b), OFormer (Li et al., 2023b), GNOT (Hao et al., 2023), and Transolver (Wu et al., 2024). Detailed dataset configurations and baseline implementations are provided in Appendix J.4.

### 5.1. Main Results: Benchmarking on Standard PDEs

The experimental results indicate that CoEvol-NO achieves the best results on the vast majority of tasks, demonstrating significant advantages in handling unstructured geometries

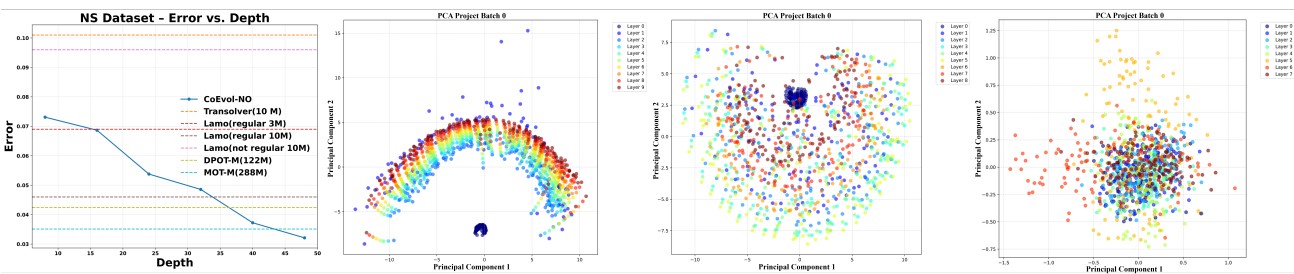

*Figure 3.* **Scaling Behavior and Latent Dynamics.** (Left) CoEvol-NO exhibits superior scaling on Navier-Stokes, outperforming much larger models (e.g., 288M MOT-M) by increasing depth. (Center-Right) PCA trajectories of three paradigms: *State-Evol* shows rigid evolution; *Co-Evol (Ours)* follows a structured expansion-contraction manifold; *Coords-Evol* displays disordered stagnation.

*Table 2.* **Main Results on Standard PDE Benchmarks.** Relative $L_2$ error is reported (lower is better). Models are categorized into: (1) **Traditional Operators**, (2) **State-Evolution** (decoupled, latent-centric), (3) **Coordinate-Evolution** (stateless, mesh-centric), and (4) **Co-Evolution** (our joint framework). Models marked with ∗ are re-implemented by us; others are cited from relevant papers (Wu et al., 2024). The best results are **bolded** and the second best are underlined.

| MODEL | POINT CLOUD | STRUCTURED MESH | | REGULAR GRID | |
|---|---|---|---|---|---|
| | ELASTICITY | AIRFOIL | PIPE | NS | DARCY |
| FNO 2020b | / | / | / | 0.1556 | 0.0108 |
| WMT 2021 | 0.0359 | 0.0075 | 0.0077 | 0.1541 | 0.0082 |
| U-FNO 2022 | 0.0239 | 0.0269 | 0.0056 | 0.2231 | 0.0183 |
| GEO-FNO 2023a | 0.0229 | 0.0138 | 0.0067 | 0.1556 | 0.0108 |
| U-NO 2022 | 0.0258 | 0.0078 | 0.0100 | 0.1713 | 0.0113 |
| F-FNO 2021 | 0.0263 | 0.0078 | 0.0070 | 0.2322 | 0.0077 |
| LSM 2023 | 0.0218 | 0.0059 | 0.0050 | 0.1535 | 0.0065 |
| GALERKIN 2021 | 0.0240 | 0.0118 | 0.0098 | 0.1401 | 0.0084 |
| HT-NET 2022 | / | 0.0065 | 0.0059 | 0.1847 | 0.0079 |
| OFORMER 2023b | 0.0183 | 0.0183 | 0.0168 | 0.1705 | 0.0124 |
| GNOT 2023 | 0.0086 | 0.0076 | 0.0047 | 0.1380 | 0.0105 |
| FACTFORMER 2023c | / | 0.0071 | 0.0060 | 0.1214 | 0.0109 |
| ONO 2023 | 0.0118 | 0.0061 | 0.0052 | 0.1195 | 0.0076 |
| REFORMER∗ 2020 | 0.0148 | 0.0310 | 0.0500 | 0.2800 | 0.0710 |
| PERCEIVER W/ CROSS∗ 2021 | 0.5553 | 0.0095 | 0.0095 | 0.2765 | 0.0150 |
| PERCEIVER W/O CROSS∗ 2021 | 0.5554 | 0.0420 | 0.0155 | 0.3443 | 0.0290 |
| TOKENLEARNER∗ 2021 | 0.2410 | 0.0756 | 0.0072 | 0.1953 | 0.0620 |
| LNO∗ 2024 | 0.3265 | 0.0196 | 0.0083 | 0.1826 | 0.0089 |
| SET TRANSFORMER∗ 2019 | 0.0322 | 0.0113 | 0.0047 | 0.3561 | 0.0086 |
| CLUSTERATTENTION∗ 2020 | 0.0057 | 0.0150 | 0.0047 | 0.0950 | 0.0067 |
| NYSTRÖMFORMER∗ 2021 | 0.0046 | 0.0074 | 0.0082 | 0.0830 | 0.0073 |
| TRANSOLVER∗ 2024 | 0.0068 | 0.0057 | 0.0039 | 0.1010 | 0.0057 |
| LAMO∗ 2025 | 0.0048 | **0.0047** | 0.0039 | 0.1176 | 0.0054 |
| **CoEvol-NO-dp** (Ours) | 0.0038 | **0.0047** | **0.0032** | **0.0731** | 0.0047 |
| **CoEvol-NO-l2** (Ours) | **0.0036** | 0.0048 | 0.0035 | 0.0904 | **0.0045** |

*Table 3.* Performance comparison on industrial benchmarks (Shape-Net Car and AirfRANS). We report the relative L2 error for Volume and Surface fields, the force coefficient ($C_D$ for Car, $C_L$ for AirfRANS), and Spearman's rank correlation ($\rho$). The best results are bolded and second best are underlined.

| MODEL | VOLUME ↓ | SURF ↓ | COEFF. $(C_D/C_L)$↓ | CORR. $(\rho)$↑ |
|---|---|---|---|---|
| *Dataset: Shape-Net Car (3D)* | | | | |
| SIMPLE MLP | 0.0512 | 0.1304 | 0.0307 | 0.9496 |
| POINTNET 2017 | 0.0494 | 0.1104 | 0.0298 | 0.9583 |
| GNOT 2023 | 0.0329 | 0.0798 | 0.0178 | 0.9833 |
| 3D-GEOCA 2024 | 0.0319 | 0.0779 | 0.0159 | 0.9842 |
| TRANSOLVER 2024 | 0.0215 | 0.0779 | 0.0125 | **0.9931** |
| CoEVOL-NO-DP | **0.0212** | 0.0699 | **0.0105** | 0.9930 |
| CoEVOL-NO-L2 | 0.0225 | **0.0693** | 0.0116 | 0.9907 |
| *Dataset: AirfRANS (2D)* | | | | |
| GRAPHSAGE 2024 | 0.0070 | 0.0161 | 0.1592 | 0.9968 |
| TRANSOLVER 2024 | 0.0088 | 0.0179 | 0.2070 | 0.9907 |
| CoEVOL-NO-DP | **0.0013** | **0.0047** | **0.0779** | **0.9990** |
| CoEVOL-NO-L2 | 0.0065 | 0.0138 | 0.2068 | 0.9942 |

**Industrial Design Tasks.** We extended our evaluation to industrial simulations using the Shape-Net Car and AirfRANS benchmarks. Table 3 details the performance across field reconstruction and coefficient estimation metrics.

In the Shape-Net Car task, the variants of CoEvol-NO exhibit specialized strengths. CoEVOL-NO-DP yields the most accurate predictions for the surrounding velocity field and drag coefficient, recording errors of 0.0212 and 0.0105, respectively. Conversely, CoEVOL-NO-L2 demonstrates superior performance in recovering surface pressure distributions, achieving an error of 0.0693, which compares favorably to the 0.0779 reported by Transolver.

For the AirfRANS, CoEVOL-NO-DP consistently surpasses the baseline methods. Notably, it lowers the volume and surface errors to 0.0013 and 0.0047, respectively, representing a distinct improvement over Transolver's 0.0088 and 0.0179. Furthermore, it achieves high precision on design-critical metrics, with a lift coefficient error of 0.0779 and a Spearman's rank correlation of 0.9990.

and dynamic evolution problems. Standard deviation results for all experiments are provided in Appendix I

Specifically, on the Elasticity benchmark (unstructured point clouds), our CoEVOL-NO-L2 variant reduces the relative error to 0.0039, an improvement of nearly 40% compared to the previous best model, Transolver (0.0064). This proves that CoEvol-NO can fully utilize the positional information of irregular meshes. Similarly, on the Navier-Stokes fluid dynamics task, CoEVOL-NO-DP achieves a record-low error of 0.0731, significantly outperforming Transolver (0.1010), which validates the effectiveness of CoEvol-NO in capturing time-evolving non-stationary physical fields. Furthermore, on regular geometry tasks such as Pipe, CoEvol-NO maintains highly competitive performance, demonstrating its robustness across different geometric complexities.

*Table 4.* Ablation study. Dual (Ours) performs most robust across both dynamic (NS) and static tasks. Latent fails on complex geometries. Sequence performs well on geometry-dominated tasks (AirfRANS) but generalizes poorly on dynamic evolution (NS).

| MODEL PARADIGM | STANDARD BENCHMARKS (REL. L2) | | | | | AIRFRANS | | | |
|---|---|---|---|---|---|---|---|---|---|
| | ELA | NS | DARCY | AIRFOIL | PIPE | VOL↓ | SURF↓ | $C_L$↓ | $\rho_L$↑ |
| CoEVOL-NO-STATES | 0.3278 | 0.1800 | 0.0083 | 0.0069 | 0.0103 | 0.1883 | 0.3850 | 0.2168 | 0.9873 |
| CoEVOL-NO-COORDS | 0.0041 | 0.1200 | 0.0078 | 0.0079 | 0.0043 | **0.0010** | **0.0032** | **0.0524** | **0.9996** |
| CoEvol-NO-Dual-dp | **0.0038** | **0.0731** | **0.0047** | **0.0048** | **0.0032** | 0.0013 | 0.0047 | 0.0779 | 0.9990 |

### 5.1.1. ABLATION STUDY I: EFFECTIVENESS OF CO-EVOLUTIONARY PARADIGMS

To validate the necessity of our Dual Co-evolutionary Framework, we benchmark CoEvol-NO against two degraded architectural paradigms: CoEvol-NO-State, which encodes the input once and evolves only the latent state without updating the mesh representation; and CoEvol-NO-Coord, which re-computes the state from the mesh at every layer without maintaining a persistent state connection.

The results in Table 4 reveal distinct characteristics of each paradigm determined by the physics of the task. First, the Latent paradigm suffers catastrophic degradation on steady-state tasks with irregular meshes, most notably on Elasticity (Error 0.3278). This failure indicates that by decoupling the state evolution from the mesh, the model loses critical positional information required to reconstruct complex boundary conditions. While efficient, the decoupled paradigm struggles to compensate for the loss of spatial constraints.

Second, the comparison between Sequence and Dual highlights a trade-off between geometric sensitivity and dynamic memory. The Sequence paradigm performs remarkably well on geometry-dominated steady-state tasks, such as Elasticity and AirfRANS. By re-encoding $\mathbf{S}$ from $\mathbf{X}$ at every layer, it repeatedly reinforces geometric information, which proves sufficient for static equilibrium problems. However, this "Coords-Evol" design falters on unsteady, time-dependent tasks. On Navier-Stokes (NS), the Sequence model underperforms the Dual model (0.12 vs. 0.071). This confirms that without a Persistent State $\mathbf{S}$, the model cannot effectively capture long-term dynamic evolution.

Consequently, the CoEvol-NO-Dual framework emerges as the most universal solution. By continuously interacting with $\mathbf{X}$, it leverages positional updates to handle complex boundaries (matching the Sequence paradigm on AirfRANS and Elasticity). Simultaneously, by maintaining a persistent evolving state $\mathbf{S}$, it captures the underlying physical dynamics required for unsteady problems.

### 5.1.2. ABLATION STUDY II: PC FRAMEWORK ON STATE

This section examines the impact of the Predictor-Corrector mechanism specifically on the evolution of the **latent state** $\mathbf{S}$. We compare the CoEvol-NO's **Exact Gradient Update** against a **First-Order Approximation** (equivalent to a standard residual update). As the compact dimension of $\mathbf{S}$ ($M \ll N$) permits more complex operations than the mesh $\mathbf{X}$, we verify whether the inclusion of the exact Jacobian term in Eq. (14) improves modeling precision.

The results confirm the necessity and superiority of explicitly utilizing the exact gradient for state correction. As shown in Table 5, for Navier-Stokes, which involves strong non-linear temporal evolution, the CoEvol-NO employing the exact gradient achieves a relative error of 0.0731, significantly outperforming the corresponding first-order approximation version (0.0830). This substantial gap indicates that simple first-order approximations ignore the local curvature information of the predictor, whereas the exact gradient, by explicitly calculating the gradient, captures the transient change directions of the physical system more precisely. A similar advantage is observed in the Elasticity task and AirFoil task, where the exact gradient update outperforms the approximation with an error of 0.0038 versus 0.0046. This further demonstrates that under the complex geometric features of unstructured meshes, the explicit error minimization strategy based on the Predictor-Corrector mechanism yields a more accurate physical operator representation than relying solely on first-order approximations. These findings suggest that the PC framework, by introducing the exact gradient, effectively overcomes the limitations of traditional approximation methods in modeling complex physical fields.

We attribute the superior performance of the Exact Gradient update to its local stability properties. Analyzing the linearized dynamics around a fixed point $\mathbf{S}^*$, the update map for the First-Order Approximation is governed by the Jacobian $\mathbf{M}_{\text{FOA}} = \mathbf{I} - \eta(\mathbf{I} - \mathbf{J})$. If the predictor's Jacobian $\mathbf{J}$ contains large imaginary eigenvalues, $\mathbf{M}_{\text{FOA}}$ can easily violate the stability condition (spectral radius $\rho < 1$).

In contrast, the Exact Gradient update is governed by $\mathbf{M}_{\text{EG}} = \mathbf{I} - \eta(\mathbf{I} - \mathbf{J})^\top(\mathbf{I} - \mathbf{J})$. This structure essentially symmetrizes the error dynamics. Since $(\mathbf{I} - \mathbf{J})^\top(\mathbf{I} - \mathbf{J})$ is positive semi-definite, its eigenvalues are real and non-negative, determined solely by the singular values of $(\mathbf{I} - \mathbf{J})$. This effectively eliminates the unstable rotational components (imaginary parts) introduced by $\mathbf{J}$. More details are provided in Appendix D

*Table 5.* Ablation study on the update mechanism. We compare the Exact Gradient (CoEvol-NO) against the First-Order Approximation (Approximation) across various benchmarks. dp: Dot-Product loss assumption (residual-style); l2: Squared loss assumption (direct-style).

| MODEL | STANDARD BENCHMARKS (REL. L2) | | | | | SHAPE-NET CAR (3D) | | | | AIRFRANS (2D) | | | |
|---|---|---|---|---|---|---|---|---|---|---|---|---|---|
| | ELA | NS | DARCY | AIRFOIL | PIPE | VOL↓ | SURF↓ | $C_D \downarrow$ | $\rho_D \uparrow$ | VOL↓ | SURF↓ | $C_L \downarrow$ | $\rho_L \uparrow$ |
| *Exact Gradient* | | | | | | | | | | | | | |
| CoEvol-NO-dp | 0.0038 | **0.0731** | 0.0047 | **0.0047** | **0.0032** | **0.0212** | 0.0699 | **0.0105** | **0.9930** | **0.0013** | **0.0047** | 0.0779 | **0.9990** |
| CoEvol-NO-l2 | **0.0036** | 0.0904 | **0.0045** | 0.0048 | 0.0035 | 0.0225 | **0.0693** | 0.0116 | 0.9907 | 0.0066 | 0.0138 | 0.2068 | 0.9942 |
| *First-Order Approximation* | | | | | | | | | | | | | |
| Approximation-dp | 0.0046 | 0.0830 | 0.0067 | 0.0057 | 0.0033 | 0.0235 | 0.0719 | 0.0121 | 0.9907 | 0.0016 | 0.0061 | **0.0709** | 0.9986 |
| Approximation-l2 | 0.0043 | 0.1100 | 0.0060 | 0.0058 | 0.0050 | 0.0230 | 0.0720 | 0.0123 | 0.9911 | 0.0018 | **0.0047** | 0.1044 | 0.9983 |

*Table 6.* Zero-shot super-resolution on Darcy flow. Models are trained on $43 \times 43$ resolution and tested on higher resolutions directly. Relative L2 errors are reported.

| MODEL | $61 \times 61$ | $85 \times 85$ | $141 \times 141$ | $211 \times 211$ | $421 \times 421$ |
|---|---|---|---|---|---|
| FNO | 0.1164 | 0.1797 | 0.2679 | 0.3160 | 0.3631 |
| ONO | 0.0204 | 0.0259 | 0.0315 | 0.0349 | 0.0386 |
| GNOT | 0.0256 | 0.0275 | 0.0294 | 0.0305 | 0.0317 |
| TRANSOLVER | 0.0239 | 0.0240 | 0.0258 | 0.0279 | 0.0290 |
| LNO | 0.0179 | 0.0192 | 0.0214 | 0.0229 | 0.0244 |
| CoEvol-NO-dp | **0.0125** | **0.0131** | **0.0161** | **0.0174** | **0.0187** |

*Table 7.* Out-of-Distribution (OOD) Generalization on AirfRANS. We evaluate the models on unseen Reynolds numbers and Angles of Attack (AoA). CoEvol-NO demonstrates superior ranking capability (higher $\rho_L$) across both settings and better volumetric generalization on the AoA task. Best results are bolded.

| MODEL | VOL ↓ | SURF ↓ | $C_L \downarrow$ | $\rho_L \uparrow$ |
|---|---|---|---|---|
| *Setting 1: OOD Reynolds* | | | | |
| TRANSOLVER | **0.0099** | **0.0451** | **0.2459** | 0.9824 |
| COEVOL-NO-DP | 0.0106 | 0.0475 | 0.2903 | **0.9933** |
| *Setting 2: OOD Angle of Attack (AoA)* | | | | |
| TRANSOLVER | 0.0363 | **0.2205** | **0.1091** | 0.9949 |
| COEVOL-NO-DP | **0.0293** | 0.2711 | 0.1437 | **0.9950** |

## 5.2. Further Studies

**Zero-Shot Super-Resolution and Out-of-Distribution Generalization** We evaluate the generalization capabilities of CoEvol-NO in terms of zero-shot super-resolution and out-of-distribution (OOD) performance.

As shown in Table 6, CoEvol-NO achieves the lowest error across all resolutions on the Darcy dataset. Notably, at the highest resolution ($421 \times 421$), it achieves a relative error of 0.0187, surpassing the second-best LNO (0.0244). This demonstrates the discretization invariance of the model.

Table 7 presents the results on AirfRANS. While Transolver retains an advantage in certain L2 error metrics, CoEvol-NO demonstrates competitive robustness in other key aspects. CoEvol-NO achieves higher Spearman's rank correlations ($\rho_L$) than Transolver in both Reynolds and AoA settings. Furthermore, in the AoA task, CoEvol-NO yields a lower

volume error compared to Transolver (0.0293 vs. 0.0363). These results indicate that CoEvol-NO maintains stable performance when generalizing to unseen physical conditions.

**Case Study and Scaling Analysis** We interpret the internal mechanism and scaling behavior of CoEvol-NO in Fig. 3. The scaling plot (left) shows that increasing depth leads to a consistent monotonic error decay on the Navier-Stokes task. Notably, our 10M model outperforms significantly larger architectures such as MOT-M (288M) (Wang et al., 2025a) and DPOT-M (122M) (Hao et al., 2024).

PCA projections (center-right) reveal distinct evolutionary traits across different paradigms. **Coords-Evol** exhibits highly overlapping states across layers; because the state is reset as fixed learnable parameters at each layer without an evolution mechanism, it fails to accumulate progressive physical memory. While **State-Evol** shows a clear layered structure, its states are clustered excessively tightly, indicating a slow evolution rate that hinders convergence efficiency. In contrast, our **Co-Evol** paradigm follows a structured "expansion-then-contraction" trajectory. As investigated in App. J, CoEvol-NO maintains high consistency across different samples whereas State-Evol displays significant variance. This stability suggests that CoEvol-NObetter captures the intrinsic physical laws through a robust optimization path, while the high variance in State-Evol indicates a failure to converge to a unified underlying operator.

## 5.3. Generality of the Predictor-Corrector Framework

A natural question is whether the PC framework is specific to the Cross-Attention predictor or the co-evolutionary paradigm of CoEvol-NO. We investigate this by applying the PC mechanism to two fundamentally different architectures: Transolver, which follows the Coords-Evolution paradigm; and LNO, which follows the State-Evolution paradigm with Self-Attention.

**PC on Transolver.** We apply the PC correction to Transolver's attention module, its FFN module, and both simultaneously. Table 8 reports the results. PC improves Transolver by 8–25% across all datasets. Notably, applying PC to the FFN module yields improvements comparable to applying

*Table 8.* **Cross-architecture transfer of the PC framework.** We apply PC to Transolver (Coords-Evol, Attention + MLP) and LNO (State-Evol, Self-Attention). Relative L2 errors reported. †: results from the State-Evolution variant of CoEvol-NO without X update.

| ARCHITECTURE | PC TARGET | AIRFOIL | PIPE | DARCY |
|---|---|---|---|---|
| TRANSOLVER | NONE (BASELINE) | 0.0057 | 0.0039 | 0.0057 |
| TRANSOLVER + PC | ATTENTION | 0.0045 | 0.0036 | 0.0049 |
| TRANSOLVER + PC | FFN (MLP) | 0.0050 | 0.0036 | 0.0048 |
| TRANSOLVER + PC | BOTH | **0.0043** | **0.0036** | **0.0043** |
| | | ELASTICITY | AIRFOIL | PIPE |
| LNO | NONE (BASELINE) | 0.3565 | 0.0196 | 0.0131 |
| LNO + PC | SELF-ATTENTION | 0.3342 | **0.0082** | 0.0141 |
| CoEvol-NO-LATENT† | W/O PC | 0.3357 | 0.0223 | 0.0182 |
| CoEvol-NO-LATENT† | +PC | 0.3278 | **0.0069** | 0.0103 |

it to attention (Darcy: $0.0057 \rightarrow 0.0048$ vs. $0.0049$), and combining both achieves the best result (Darcy: $0.0057 \rightarrow 0.0043$, 25% improvement).

**PC on LNO.** We apply the PC correction to LNO's Self-Attention (not Cross-Attention). PC-LNO improves Airfoil error from 0.0196 to 0.0082 (58% reduction), while Elasticity remains high ($0.3565 \rightarrow 0.3342$) due to the architectural limitation of lacking an X update—confirming that PC enhances the *update mechanism* but cannot compensate for missing geometric feedback. Comparing CoEvol-NO-Latent with and without PC further validates this: removing PC degrades Airfoil from 0.0069 to 0.0223, confirming PC's contribution even in the State-Evolution paradigm.

These results demonstrate that the PC framework transfers across architectures (LNO $\rightarrow$ Transolver), module types (Self-Attention $\rightarrow$ Cross-Attention $\rightarrow$ MLP), and evolution paradigms (State-Evolution $\rightarrow$ Coords-Evolution), suggesting it is a general principle for improving layer-wise updates in neural operator transformers.

### 5.4. Ablation: Momentum and LayerScale

The PC correction introduces two critical components: layer-wise momentum (Eq. (12)) and learnable step sizes via LayerScale (Eq. (13)). We ablate each to understand their contributions.

**Momentum.** We vary $\beta_S$ and $\beta_X$ independently. Table 9 reveals a striking asymmetry: $S$-momentum ($\beta_S = 0.9$) yields substantial improvement (Pipe: $-32\%$, Airfoil: $-20\%$), while $X$-momentum provides negligible benefit. This is because the exact gradient is computed independently at each layer, producing inconsistent correction directions; momentum accumulates gradient history for smoother updates. For $X$, the first-order update lacks this structure, so momentum adds no benefit.

**LayerScale.** Table 10 compares learnable LayerScale against fixed step sizes. LayerScale improves 31% on Airfoil and 54% on Pipe over the best fixed-scale alternative. Applying it to Transolver yields far less benefit (0.011 vs.

*Table 9.* **Momentum ablation.** $S$-momentum is critical; $X$-momentum provides no benefit. Full results in Appendix F.

| $\beta_S$ | $\beta_X$ | PIPE | AIRFOIL | DARCY |
|---|---|---|---|---|
| 0.0 | 0.0 | 0.0044 | 0.0059 | **0.0046** |
| 0.5 | 0.0 | 0.0036 | 0.0060 | 0.0050 |
| **0.9** | **0.0** | **0.0032** | **0.0047** | 0.0047 |
| 0.0 | 0.5 | 0.0038 | 0.0057 | 0.0048 |
| 0.0 | 0.9 | 0.0048 | 0.0050 | 0.0049 |

*Table 10.* **LayerScale ablation.** Learnable per-dimension step sizes outperform all fixed-scale alternatives.

| SETTING | AIRFOIL | PIPE |
|---|---|---|
| CONSTANT SCALE $= 1$ | 0.0068 | 0.0069 |
| CONSTANT SCALE $= 10^{-5}$ | 0.0076 | 0.0045 |
| **LAYERSCALE (LEARNABLE)** | **0.0047** | **0.0032** |
| TRANSOLVER + LAYERSCALE | 0.011 | 0.0098 |

0.0047), confirming that adaptive step sizing requires the PC's structured gradient.

## 6. Conclusion and Limitation

We present CoEvol-NO, a linear-complexity neural operator that synchronizes latent state and mesh updates via a co-evolutionary framework. This design addresses the limitations of decoupled paradigms in utilizing geometric information and Coords-Evol paradigms in retaining dynamic memory. Its core Predictor-Corrector mechanism employs explicit error-driven updates to adapt to complex physical dynamics. Experiments on five standard and two industrial benchmarks show state-of-the-art performance. With linear complexity and training stability, CoEvol-NO can serve as a backbone for large-scale physics models. Future work will explore pre-training on larger-scale datasets.

**Limitations.** We acknowledge three limitations. **(1)** The PC correction requires the predictor's Jacobian. While Cross-Attention admits a closed-form solution, most architectures (GNN, convolution) must rely on backpropagation, adding computational cost; efficient gradient computation for general modules remains open. **(2)** As shown in our ablation study (Table 11), applying PC to all modules simultaneously can be counterproductive (Pipe: $0.0033 \rightarrow 0.0049$ when all use exact gradient). A principled criterion for selective application is still lacking. **(3)** The exact gradient update can introduce conditioning challenges depending on the choice of correction loss. We mitigate this with momentum and LayerScale, but the interaction between loss geometry and optimization dynamics requires further study for extreme depth ($>40$ layers).

## Impact Statement

This paper presents work whose goal is to advance the field of Machine Learning. There are many potential societal consequences of our work, none which we feel must be specifically highlighted here.

## Acknowledgments

This work was supported by the Scientific Research Innovation Capability Support Project for Young Faculty (U40) of the Ministry of Education of China (Grant No. SRICSPYF-ZY2025019).

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

# A. Theoretical Analysis and Derivations

## A.1. Analytical Derivation of the Exact Gradient for CoEvol-NO

In this section, we provide the rigorous step-by-step derivation of the exact gradient for the Predictor-Corrector update mechanism, corresponding to Eq. (14) in the main text. We employ matrix calculus and the properties of the Trace operator to derive the Jacobian-vector product efficiently.

### A.1.1. PROBLEM SETUP

Let $\mathbf{S} \in \mathbb{R}^{M \times C}$ denote the latent state and $C(\mathbf{S})$ denote the predictor function (Cross-Attention) that generates the target estimate. The correction objective is defined as the squared Frobenius norm of the error:

$$\mathcal{L}(\mathbf{S}) = \frac{1}{2}\|\mathbf{S} - C(\mathbf{S})\|_F^2 = \frac{1}{2}\mathrm{Tr}\left((\mathbf{S} - C(\mathbf{S}))^\top (\mathbf{S} - C(\mathbf{S}))\right) \tag{15}$$

The predictor $C(\mathbf{S})$ is instantiated as a Cross-Attention module where $\mathbf{S}$ acts as the Query:

$$C(\mathbf{S}) = \mathrm{softmax}\left(\frac{\mathbf{SW}_Q(\mathbf{XW}_K)^\top}{\sqrt{d_k}}\right)(\mathbf{XW}_V) \tag{16}$$

To simplify notation, let $\mathbf{Q} = \mathbf{SW}_Q$, $\mathbf{K} = \mathbf{XW}_K$, $\mathbf{V} = \mathbf{XW}_V$, and $\mathbf{A} = \frac{\mathbf{QK}^\top}{\sqrt{d_k}}$. The attention map is $\mathbf{P} = \mathrm{softmax}(\mathbf{A})$, thus $C(\mathbf{S}) = \mathbf{PV}$.

### A.1.2. GRADIENT DECOMPOSITION

We seek the total gradient $d\mathcal{L}$ with respect to a variation $d\mathbf{S}$. Applying the differential operator $d(\cdot)$ to the loss:

$$d\mathcal{L} = \frac{1}{2}\mathrm{Tr}\left(d\left((\mathbf{S} - C)^\top(\mathbf{S} - C)\right)\right) \tag{17}$$

$$= \mathrm{Tr}\left((\mathbf{S} - C)^\top d(\mathbf{S} - C)\right) \tag{18}$$

$$= \mathrm{Tr}\left((\mathbf{S} - C)^\top (d\mathbf{S} - dC)\right) \tag{19}$$

$$= \underbrace{\mathrm{Tr}\left((\mathbf{S} - C)^\top d\mathbf{S}\right)}_{\text{Direct Term}} - \underbrace{\mathrm{Tr}\left((\mathbf{S} - C)^\top dC\right)}_{\text{Jacobian Term}} \tag{20}$$

Let $\mathbf{\Delta} = \mathbf{S} - C(\mathbf{S})$ denote the residual error. The first term yields the direct gradient $\mathbf{\Delta}$. The second term requires backpropagating the error $-\mathbf{\Delta}$ through the predictor $C(\mathbf{S})$. Let $\mathbf{G}_{out} = -\mathbf{\Delta} = -(\mathbf{S} - C(\mathbf{S}))$. We now derive the gradient of the trace term $\mathrm{Tr}(\mathbf{G}_{out}^\top dC)$ through the attention mechanism.

### A.1.3. BACKPROPAGATION THROUGH ATTENTION

1. Gradient w.r.t Attention Map $\mathbf{P}$: Since $C = \mathbf{PV}$ and $\mathbf{V}$ is independent of $\mathbf{S}$:

$$\mathrm{Tr}(\mathbf{G}_{out}^\top dC) = \mathrm{Tr}(\mathbf{G}_{out}^\top(d\mathbf{PV})) = \mathrm{Tr}((\mathbf{G}_{out}\mathbf{V}^\top)^\top d\mathbf{P}) \tag{21}$$

Thus, the gradient at $\mathbf{P}$ is $\mathbf{G}_P = \mathbf{G}_{out}\mathbf{V}^\top = -(\mathbf{S} - C(\mathbf{S}))\mathbf{V}^\top$.

2. Gradient w.r.t Pre-softmax Logits $\mathbf{A}$: The relation is $\mathbf{P} = \mathrm{softmax}(\mathbf{A})$. For the softmax function, the differential relationship is given by:

$$d\mathcal{L} = \mathrm{Tr}(\mathbf{G}_P^\top d\mathbf{P}) = \mathrm{Tr}(\mathbf{G}_A^\top d\mathbf{A}) \tag{22}$$

Considering the element-wise derivative of softmax, we derive:

$$dP_{ij} = P_{ij}dA_{ij} - P_{ij}\sum_k P_{ik}dA_{ik} \tag{23}$$

Substituting this into the trace formulation:

$$\text{Tr}(\mathbf{G}_P^\top d\mathbf{P}) = \sum_{i,j}(\mathbf{G}_P)_{ij}dP_{ij} \tag{24}$$

$$= \sum_{i,j}(\mathbf{G}_P)_{ij}\left(P_{ij}dA_{ij} - P_{ij}\sum_k P_{ik}dA_{ik}\right) \tag{25}$$

$$= \sum_{i,j}((\mathbf{G}_P)_{ij}P_{ij})dA_{ij} - \sum_i\left(\sum_j(\mathbf{G}_P)_{ij}P_{ij}\right)\sum_k P_{ik}dA_{ik} \tag{26}$$

Let $\lambda = \text{rowsum}(\mathbf{G}_P \circ \mathbf{P}) \in \mathbb{R}^{M \times 1}$. The second term can be written in matrix form using the broadcasting of $\lambda$ (denoted as $\lambda \cdot \mathbf{1}^\top$). The gradient $\mathbf{G}_A$ is thus:

$$\mathbf{G}_A = \mathbf{P} \circ \mathbf{G}_P - \mathbf{P} \circ (\text{rowsum}(\mathbf{G}_P \circ \mathbf{P}) \cdot \mathbf{1}^\top) \tag{27}$$

where $\circ$ denotes the Hadamard product.

3. Gradient w.r.t Latent State $\mathbf{S}$: Finally, backpropagating through the linear projection $\mathbf{A} = \frac{\mathbf{SW}_Q\mathbf{K}^\top}{\sqrt{d_k}}$:

$$\text{Tr}(\mathbf{G}_A^\top d\mathbf{A}) = \text{Tr}\left(\mathbf{G}_A^\top \frac{d\mathbf{SW}_Q\mathbf{K}^\top}{\sqrt{d_k}}\right) = \frac{1}{\sqrt{d_k}}\text{Tr}\left((\mathbf{G}_A\mathbf{KW}_Q^\top)^\top d\mathbf{S}\right) \tag{28}$$

Thus, the Jacobian-vector product component is $\mathbf{G}_S = \frac{1}{\sqrt{d_k}}\mathbf{G}_A\mathbf{KW}_Q^\top$.

### A.1.4. FINAL EXACT GRADIENT FORMULA

Combining the direct term and the Jacobian correction term, we obtain the final closed-form gradient update rule used in CoEvol-NO:

$$\nabla_\mathbf{S}\mathcal{L} = (\mathbf{S} - C(\mathbf{S})) + \mathbf{G}_S \tag{29}$$

Substituting the intermediate terms:

$$\nabla_\mathbf{S}\mathcal{L} = (\mathbf{S} - \mathbf{PV}) - \frac{1}{\sqrt{d_k}}\left[(\mathbf{\Gamma} - \text{rowsum}(\mathbf{\Gamma} \circ \mathbf{P})\mathbf{1}^\top) \circ \mathbf{P}\right]\mathbf{KW}_Q^\top \tag{30}$$

where $\mathbf{\Gamma} = (\mathbf{S} - \mathbf{PV})\mathbf{V}^\top$ represents the error projected onto the value space. This analytical form allows CoEvol-NO to update the state along the exact error landscape curvature, capturing high-order dependencies ignored by first-order approximations.

### A.2. Analytical Derivation for Mesh Sequence X under Dot-Product Loss

While the latent state $\mathbf{S}$ is updated via squared error, we consider the case where the mesh sequence $\mathbf{X} \in \mathbb{R}^{N \times C}$ is optimized to maximize the alignment with the predicted features, corresponding to the Dot-Product Loss:

$$\mathcal{L}(\mathbf{X}) = -\text{Tr}\left(\mathbf{X}^\top C(\mathbf{X})\right) \tag{31}$$

Here, the predictor $C(\mathbf{X})$ uses $\mathbf{X}$ as the Query and the updated latent state $\mathbf{S}$ as Key and Value:

$$C(\mathbf{X}) = \text{softmax}\left(\frac{\mathbf{XW}_Q(\mathbf{SW}_K)^\top}{\sqrt{d_k}}\right)(\mathbf{SW}_V) = \mathbf{PV}_s \tag{32}$$

where $\mathbf{P} \in \mathbb{R}^{N \times M}$ and $\mathbf{V}_s = \mathbf{SW}_V \in \mathbb{R}^{M \times C}$.

### A.2.1. DERIVATION OF THE EXACT GRADIENT

Applying the differential operator $d(\cdot)$ to the dot-product objective:

$$d\mathcal{L} = -\text{Tr}(d\mathbf{X}^\top C(\mathbf{X})) - \text{Tr}(\mathbf{X}^\top dC(\mathbf{X})) \tag{33}$$

$$= \underbrace{-\text{Tr}(C(\mathbf{X})^\top d\mathbf{X})}_{\text{Direct Term}} - \underbrace{\text{Tr}(\mathbf{X}^\top d\mathbf{PV}_s)}_{\text{Jacobian Term}} \tag{34}$$

The first term yields the first-order gradient component $-C(\mathbf{X})$. For the second term, since $\mathbf{V}_s$ is independent of $\mathbf{X}$ during this update step, we define the intermediate gradient $\mathbf{G}_P = -\mathbf{X}\mathbf{V}_s^\top \in \mathbb{R}^{N \times M}$. Following the same softmax chain rule derivation as in Appendix A.1.3:

$$\mathbf{G}_A = \mathbf{P} \circ \mathbf{G}_P - \mathbf{P} \circ (\text{rowsum}(\mathbf{G}_P \circ \mathbf{P}) \cdot \mathbf{1}_M^\top) \tag{35}$$

Backpropagating to $\mathbf{X}$ through the query projection $\mathbf{Q}_x = \mathbf{X}\mathbf{W}_Q$:

$$\mathbf{G}_X = \frac{1}{\sqrt{d_k}}\mathbf{G}_A(\mathbf{S}\mathbf{W}_K)\mathbf{W}_Q^\top \tag{36}$$

The final exact gradient for $\mathbf{X}$ under dot-product loss is:

$$\nabla_{\mathbf{X}}\mathcal{L} = -C(\mathbf{X}) + \mathbf{G}_X \tag{37}$$

### A.2.2. APPROXIMATION RATIONALE FOR THE MESH SEQUENCE

As derived above, the exact gradient for $\mathbf{X}$ is analytically available. Our 8-combination ablation (Table 11) confirms that applying PC to $\mathbf{X}$ alone yields comparable gains to applying PC to $\mathbf{S}$ alone (Airfoil: $0.0057 \to 0.0048$ in both cases), and applying PC to both can be effective in certain settings (e.g., Transolver with both PC: Darcy $0.0057 \to 0.0043$). From a computational perspective, the end-to-end overhead of $\mathbf{X}$-PC is only approximately 16% higher than $\mathbf{S}$-PC (Table 19), confirming that cost alone does not explain the design choice.

Our default configuration applies the PC exact gradient to the latent state $\mathbf{S}$ while adopting a first-order approximation for $\mathbf{X}$. This choice is currently empirical rather than theoretically grounded. A principled criterion for determining which components of a model should adopt the PC framework—based on, for instance, information bottleneck principles, gradient flow analysis, or learned adaptive mechanisms that dynamically assign PC updates to the most beneficial submodules—remains an open question and an important direction for future work.

## B. Theoretical Motivation for the Prediction Target Selection

In the CoEvol-NO framework, the update rule is formalized as an optimization step minimizing a local objective $\mathcal{L}(\mathbf{S}_n, \mathbf{S}^*)$, where $\mathbf{S}^*$ acts as a proxy for the ideal next state. A critical design choice lies in the definition of this approximation target $\mathbf{S}^*$. Drawing analogies from numerical methods, standard neural network updates—such as $S_{n+1} = f(S_n)$ (Direct Substitution) or $S_{n+1} = S_n + f(S_n)$ (Residual Connection)—can be viewed as explicit prediction steps. In this section, we analyze two distinct formulations for the prediction target $\hat{\mathbf{S}}$ and justify our selection of the *Direct Mapping* target.

### B.1. Analysis of Potential Prediction Targets

We consider the layer-wise evolution as a dynamic process where a predictor $f(\cdot)$ proposes a transition. We contrast two paradigms: Case A (Direct Mapping Target), where the target is defined as $\hat{\mathbf{S}} = f(\mathbf{S}_n)$, implying the state should move towards the predictor's output; and Case B (Accumulative Target), where $\hat{\mathbf{S}} = \mathbf{S}_n + f(\mathbf{S}_n)$, implying the target is the result of a residual addition.

**Case A: The Direct Mapping Formulation (Adopted).** When we define the target as $\hat{\mathbf{S}} = f(\mathbf{S})$, we essentially treat the predictor as an operator that outputs the desired state directly. Under a Squared L2 Loss assumption, the objective becomes $\mathcal{L} = \frac{1}{2}\|\mathbf{S} - f(\mathbf{S})\|^2$. Letting the residual be $\mathbf{r} = \mathbf{S} - f(\mathbf{S})$, the gradient is derived as $\nabla_{\mathbf{S}}\mathcal{L} = (\mathbf{I} - \mathbf{J})^\top(\mathbf{S} - f(\mathbf{S}))$, where $\mathbf{J}$ is the Jacobian of $f$. The resulting update direction is $-\eta(\mathbf{I} - \mathbf{J})^\top(\mathbf{S} - f(\mathbf{S}))$. This form explicitly utilizes the mismatch between the current state and the predicted state to drive the update, corrected by the Jacobian term $(\mathbf{I} - \mathbf{J})^\top$ which accounts for the local curvature of the predictor. Alternatively, under a Dot Product Loss $\mathcal{L} = -\langle\mathbf{S}, f(\mathbf{S})\rangle$, the gradient becomes $-(f(\mathbf{S}) + \mathbf{J}^\top\mathbf{S})$, leading to an update direction $\eta(f(\mathbf{S}) + \mathbf{J}^\top\mathbf{S})$.

**Case B: The Accumulative Target Formulation.** Defining the target as $\hat{\mathbf{S}} = \mathbf{S} + f(\mathbf{S})$ reveals interesting theoretical properties, although distinct from our desired behavior. Under a Squared L2 Loss, the objective simplifies to $\mathcal{L} = \frac{1}{2}\|\mathbf{S} - (\mathbf{S} + f(\mathbf{S}))\|^2 = \frac{1}{2}\|f(\mathbf{S})\|^2$. The gradient $\nabla_{\mathbf{S}}\mathcal{L} = \mathbf{J}^\top f(\mathbf{S})$ implies that the optimization goal is to minimize the magnitude of the update $f(\mathbf{S})$ itself. This can be interpreted as an *Implicit Kinetic Energy Minimization*, where

the system encourages stability by penalizing large state transitions. Under a Dot Product Loss, the objective becomes $\mathcal{L} = -\langle \mathbf{S}, \mathbf{S} + f(\mathbf{S}) \rangle = -\|\mathbf{S}\|^2 - \langle \mathbf{S}, f(\mathbf{S}) \rangle$. Here, a crucial phenomenon emerges: the term $-\|\mathbf{S}\|^2$ appears naturally. This acts as an *Intrinsic Norm Regularization*, functionally equivalent to Weight Decay, which prevents the unbounded growth of the latent state magnitude without requiring external regularization hyperparameters.

### B.2. Justification for Choosing Case A

While Case B offers intriguing properties such as intrinsic regularization, we adopt Case A ($\hat{\mathbf{S}} = f(\mathbf{S})$) for CoEvol-NO based on the role of the latent state as an information bottleneck. We conceptualize the predictor $f(\mathbf{S}, \mathbf{X})$ as a *Compressor* that extracts relevant physical information from the mesh $\mathbf{X}$ and condenses it into the latent manifold $\mathbf{S}$. Under this view, an optimal state $\mathbf{S}^*$ should be a fixed point of this information extraction process, satisfying the consistency condition $\mathbf{S}^* \approx f(\mathbf{S}^*, \mathbf{X})$. Therefore, minimizing the discrepancy $\mathcal{L}(\mathbf{S}_n, f(\mathbf{S}_n))$ aligns with the goal of finding a representation that is maximally consistent with the current layer's feature extraction. The Direct Mapping target ensures that the gradient descent step actively bridges the gap between the current state and the optimal representation predicted by the mesh interaction, making it the most logical choice for our Co-Evolutionary framework. The exploration of Accumulative Targets and their intrinsic regularization properties remains a promising direction for future research.

## C. CoEvol-NO Algorithm

---

**Algorithm 1** Predictor-Corrector Co-Evolution for Neural Operators

---

**Require:** Input mesh sequence $\mathbf{X}_0$, initial latent state $\mathbf{S}_0$, number of layers $L$
**Ensure:** Output prediction $\hat{\mathbf{y}}$
1: Initialize velocity buffer $\mathbf{V}_0 \leftarrow \mathbf{0}$
2: **for** $t = 1$ to $L$ **do**
3:     {Predictor: Cross-Attention}
4:     $\hat{\mathbf{S}}_t \leftarrow \text{CrossAttention}(\mathbf{S}_{t-1}, \mathbf{X}_{t-1})$
5:     {Corrector: Exact Gradient Update for $\mathbf{S}$}
6:     $\mathbf{g}_t \leftarrow \nabla_{\mathbf{S}_{t-1}} \mathcal{L}_{\text{corr}}(\mathbf{S}_{t-1}; \hat{\mathbf{S}}_t)$ {Exact gradient of correction loss}
7:     $\mathbf{V}_t \leftarrow \beta_S \cdot \mathbf{V}_{t-1} + \mathbf{g}_t$ {Momentum accumulation}
8:     $\mathbf{S}_t \leftarrow \mathbf{S}_{t-1} - \text{LayerScale}(\mathbf{V}_t)$ {Learnable step sizes}
9:     {Mesh Update: First-Order Approximation}
10:     $\mathbf{X}_t \leftarrow \mathbf{X}_{t-1} + \Delta\mathbf{X}_t$ {Gradient-free refinement}
11: **end for**
12: $\hat{\mathbf{y}} \leftarrow \text{Decoder}(\mathbf{S}_L)\ \hat{\mathbf{y}}$

---

This algorithm formalizes the layer-wise co-evolutionary process. At each layer, the Predictor uses Cross-Attention to aggregate geometric information from $\mathbf{X}_{t-1}$ into a latent target estimate $\hat{\mathbf{S}}_t$. The Corrector then refines $\mathbf{S}_{t-1}$ via exact gradient descent on the correction loss $\mathcal{L}_{\text{corr}}(\mathbf{S}_{t-1}; \hat{\mathbf{S}}_t)$, enhanced by momentum ($\beta_S$) and LayerScale. Concurrently, the mesh sequence $\mathbf{X}_t$ is updated via a first-order approximation to preserve local details. The exact gradient derivation for the Cross-Attention predictor is provided in Appendix A.1.

## D. Stability and Conditioning Analysis of Update Mechanisms

In this section, we provide a rigorous comparative analysis between the First-Order Approximation (used in standard ResNets/RNNs) and the Exact Gradient Update (used in CoEvol-NO) regarding their local stability and optimization landscape conditioning.

### D.1. Local Stability Analysis

We analyze the discrete dynamical systems formed by the update rules around a fixed point $\mathbf{S}^*$ satisfying $\mathbf{S}^* = f(\mathbf{S}^*)$. Let $\mathbf{J} = \nabla f(\mathbf{S}^*)$ denote the Jacobian of the predictor at the fixed point. The stability is determined by the spectral radius $\rho(\cdot)$ of the Jacobian of the update mapping $\mathbf{F}(\mathbf{S})$. A system is locally stable if $\rho(\nabla \mathbf{F}(\mathbf{S}^*)) < 1$.

**1. First-Order Approximation System.**    The update rule is $\mathbf{S}_{t+1} = \mathbf{S}_t - \eta(\mathbf{S}_t - f(\mathbf{S}_t))$. The Jacobian of this mapping $\mathbf{F}_1$ is:

$$\nabla \mathbf{F}_1(\mathbf{S}^*) = \mathbf{I} - \eta(\mathbf{I} - \mathbf{J}) \tag{38}$$

The eigenvalues are $\mu_i = 1 - \eta(1 - \lambda_i(\mathbf{J}))$. For stability, all eigenvalues must lie within the unit circle in the complex plane. Instability Risk: If the physical system involves strong rotation or convection (common in fluids), $\mathbf{J}$ will have eigenvalues with large imaginary parts. In this case, $\mu_i$ can easily drift outside the unit circle, causing the dynamics to diverge even with small $\eta$.

**2. Exact Gradient System (CoEvol-NO).**    The update rule is derived from minimizing $\mathcal{L} = \frac{1}{2}\|\mathbf{S} - f(\mathbf{S})\|^2$:

$$\mathbf{S}_{t+1} = \mathbf{S}_t - \eta \left[ (\mathbf{I} - \nabla f(\mathbf{S}_t))^\top (\mathbf{S}_t - f(\mathbf{S}_t)) \right] \tag{39}$$

At the fixed point where the residual $\mathbf{S}^* - f(\mathbf{S}^*) = \mathbf{0}$, the Hessian term involving second-order derivatives vanishes. The Jacobian of this mapping $\mathbf{F}_2$ simplifies to:

$$\nabla \mathbf{F}_2(\mathbf{S}^*) = \mathbf{I} - \eta(\mathbf{I} - \mathbf{J})^\top (\mathbf{I} - \mathbf{J}) \tag{40}$$

Let $\mathbf{A} = \mathbf{I} - \mathbf{J}$. The update matrix becomes $\mathbf{I} - \eta \mathbf{A}^\top \mathbf{A}$. Since $\mathbf{A}^\top \mathbf{A}$ is symmetric positive semi-definite, its eigenvalues are $\sigma_i(\mathbf{A})^2 \geq 0$, where $\sigma_i$ are the singular values of $\mathbf{A}$. Stability Advantage: The eigenvalues of the update matrix are $1 - \eta \sigma_i(\mathbf{A})^2$, which are strictly real. By "symmetrizing" the error dynamics, the Exact Gradient mechanism effectively eliminates the unstable rotational components (imaginary parts) introduced by $\mathbf{J}$. The stability condition simplifies to $\eta < 2/\sigma_{\max}^2(\mathbf{A})$, which is a robust constraint on the step size independent of the rotational dynamics.

### D.2. Conditioning Analysis: The Price of Precision

While the Exact Gradient update offers superior stability, it introduces a challenge regarding the conditioning of the optimization landscape. We analyze the condition number $\kappa$ of the operator mapping the error signal $\delta$ to the update gradient.

**Conditioning Squaring.**    Let $\mathbf{A} = \mathbf{I} - \mathbf{J}$ be the residual operator with singular values $\sigma_1 \geq \cdots \geq \sigma_n > 0$. The base condition number is $\kappa(\mathbf{A}) = \sigma_1/\sigma_n$.

- First-Order Approximation: The update direction is $\mathbf{g}_1 \approx \mathbf{A}\delta$. The condition number of this operator is simply $\kappa_1 = \kappa(\mathbf{A})$. The difficulty of training scales linearly with the stiffness of the physical problem.

- Exact Gradient: The update direction is $\mathbf{g}_2 \approx \mathbf{A}^\top \mathbf{A}\delta$. The singular values of the operator $\mathbf{A}^\top \mathbf{A}$ are the squares of the singular values of $\mathbf{A}$. Consequently, the condition number becomes:

$$\kappa_2 = \frac{\sigma_{\max}^2(\mathbf{A})}{\sigma_{\min}^2(\mathbf{A})} = (\kappa(\mathbf{A}))^2 \tag{41}$$

### D.3. Conditioning under Dot-Product Loss.

For dot-product loss $\mathcal{L} = -\langle \mathbf{S}, f(\mathbf{S}) \rangle$, let $\mathbf{J} = \nabla_{\mathbf{S}} f(\mathbf{S})$ denote the Jacobian. The Hessian matrix is $\nabla^2 \mathcal{L} = -(\mathbf{J} + \mathbf{J}^\top)$. The condition number is

$$\kappa(\mathbf{J} + \mathbf{J}^\top) \approx \kappa(\mathbf{J}) \tag{42}$$

Unlike L2 loss which exhibits $\kappa^2$ scaling due to the Gram matrix structure $\mathbf{A}^\top \mathbf{A}$, dot-product loss preserves the base condition number without squaring.

**Implication and Solution.**    The squared condition number $\kappa_2$ implies that the optimization landscape for the Exact Gradient is significantly more ill-conditioned than the approximation. Dominant eigen-directions ($\sigma_1^2$) are amplified, while subtle physical details ($\sigma_n^2$) may be numerically annihilated.

This theoretical insight justifies our architectural choices: 1. Necessity of Momentum: We employ momentum-based updates (Eq. 10 in main text) to smooth out the oscillations caused by the high curvature ($\sigma_{\max}^2$) and accelerate convergence along shallow valleys. 2. Orthogonalization (Future Work): This analysis suggests that advanced optimizers like Muon, which perform orthogonalization to whiten the gradient, are theoretically optimal for this framework as they can counteract the conditioning squaring effect ($\kappa^2 \to \kappa$) while retaining the stability benefits of the exact gradient.

## E. Full Gradient Ablation: 8-Combination Study

We conduct a full factorial ablation across all three gradient computation points in CoEvol-NO: the $S$ predictor, the $X$ update, and the FFN module. For each, we test exact gradient vs. first-order approximation, yielding $2^3 = 8$ combinations.

*Table 11.* Full 8-combination gradient ablation. "Exact" denotes exact gradient (with Jacobian), "Appro" denotes first-order approximation. Relative L2 error reported.

| $S$ GRAD | $X$ GRAD | FFN GRAD | PIPE | AIRFOIL |
|----------|----------|----------|--------|---------|
| APPRO | APPRO | APPRO | 0.0033 | 0.0057 |
| EXACT | APPRO | APPRO | 0.0032 | **0.0047** |
| APPRO | EXACT | APPRO | 0.0033 | 0.0048 |
| EXACT | EXACT | APPRO | 0.0038 | 0.0050 |
| APPRO | APPRO | **EXACT** | **0.0026** | 0.0052 |
| EXACT | APPRO | EXACT | 0.0030 | 0.0049 |
| APPRO | EXACT | EXACT | 0.0028 | 0.0048 |
| EXACT | EXACT | EXACT | 0.0049 | 0.0050 |

Key observations: **(1)** FFN exact gradient yields the largest single improvement on Pipe ($0.0033 \to 0.0026$, 21%), suggesting PC's benefit extends beyond attention modules. **(2)** Exact $X$ gradient improves Airfoil ($0.0057 \to 0.0048$) but at significant memory cost ($\times 1.9$, see Appendix M). **(3)** Applying exact gradient everywhere is counterproductive (Pipe: $0.0033 \to 0.0049$), indicating that selective application is optimal. The default design (exact gradient for $S$, first-order for $X$ and FFN) balances accuracy and computational cost.

## F. Full Momentum Ablation

Table 12 reports the full $\beta_S \times \beta_X$ ablation across all 9 combinations on Pipe, Airfoil, and Darcy.

*Table 12.* Full momentum ablation across all $\beta_S$ and $\beta_X$ combinations. Relative L2 error reported with exact gradient on $S$ and LayerScale enabled.

| $\beta_S$ | $\beta_X$ | PIPE | AIRFOIL | DARCY |
|-----------|-----------|--------|---------|--------|
| 0.0 | 0.0 | 0.0044 | 0.0059 | 0.0046 |
| 0.5 | 0.0 | 0.0036 | 0.0060 | 0.0050 |
| **0.9** | **0.0** | **0.0032** | **0.0047** | 0.0047 |
| 0.0 | 0.5 | 0.0038 | 0.0057 | 0.0048 |
| 0.0 | 0.9 | 0.0048 | 0.0050 | 0.0049 |
| 0.5 | 0.5 | 0.0037 | 0.0058 | **0.0045** |
| 0.5 | 0.9 | 0.0037 | 0.0052 | 0.0048 |
| 0.9 | 0.5 | 0.0037 | 0.0052 | 0.0047 |
| 0.9 | 0.9 | 0.0038 | 0.0049 | 0.0052 |

**Asymmetry between $S$- and $X$-momentum.** The most striking pattern is the stark asymmetry: increasing $\beta_S$ from 0 to 0.9 (top block, $\beta_X = 0$) improves Pipe by 32% and Airfoil by 20%. In contrast, increasing $\beta_X$ from 0 to 0.9 (rows 4–5) either provides no benefit or actively degrades performance (Pipe: $0.0044 \to 0.0048$). This asymmetry is explained by the update structure: the exact gradient for $S$ is computed independently at each layer, producing correction directions that can be inconsistent across layers. Momentum accumulates gradient history via $V_t = \beta V_{t-1} + g_t$ to provide a smoother, more consistent correction direction. For $X$, the update uses only the first-order residual without the PC mechanism, so the conditioning is inherently different and momentum provides no benefit.

**Combining both momenta does not help.** The lower-right block (both $\beta_S, \beta_X > 0$) shows that $X$-momentum actively interferes with $S$-momentum: $\beta_S = 0.9, \beta_X = 0.5$ (0.0037) is worse than $\beta_S = 0.9, \beta_X = 0$ (0.0030) on Pipe. This confirms that the optimal configuration is $\beta_S = 0.9, \beta_X = 0$.

## G. Comparison with Attention Residuals (AttnRes)

We compare the PC framework with AttnRes (Team et al., 2026), which addresses the uniform dilution problem in deep networks via data-dependent scalar re-weighting $\alpha_{i \to l}$. We test AttnRes on CoEvol-NO's backbone by applying it to the X update path, the S update path, or both:

*Table 13.* AttnRes vs. PC framework on CoEvol-NO's backbone.

| ATTNRES APPLIED TO | AIRFOIL | PIPE |
|---|---|---|
| NEITHER (STANDARD RESIDUAL) | 0.0057 | 0.0033 |
| X SIDE ONLY | 0.0054 | 0.0076 |
| S SIDE ONLY | 0.0080 | 0.0065 |
| BOTH SIDES | 0.0056 | 0.0070 |
| PC FRAMEWORK (OURS) | **0.0047** | **0.0032** |

AttnRes provides marginal improvement on Airfoil when applied to the X side ($0.0057 \to 0.0054$) but catastrophically degrades Pipe ($0.0033 \to 0.0076$). Applying AttnRes to the S side or both sides consistently hurts. In contrast, the PC framework improves both tasks simultaneously. We note that AttnRes was designed primarily for very deep networks where the uniform dilution problem is most severe; our current models use moderate depth ($L = 8$), which may not fully reflect AttnRes's intended setting.

## H. Extended Scaling Analysis

In this section, we provide a detailed scaling analysis of CoEvol-NO to evaluate its performance limits and numerical stability under varying architectural configurations. We focus on two key hyperparameters: the model depth (number of layers $L$) and the latent dimension $M$.

### H.1. Scaling with Model Depth

To assess whether CoEvol-NO benefits from increased depth like traditional deep learning backbones, we vary the number of Predictor-Corrector blocks $L \in \{8, 16, 24, 32, 40\}$. The relative $L_2$ errors across five standard PDE benchmarks are reported in Table 14.

*Table 14.* **Scaling Analysis of Model Depth.** Performance of CoEvol-NO as a function of the number of layers $L$. Results demonstrate significant scaling gains especially for non-stationary tasks like Navier-Stokes.

| DEPTH ($L$) | PIPE | ELASTICITY | AIRFOIL | DARCY | NAVIER-STOKES |
|---|---|---|---|---|---|
| $L = 8$ (DEFAULT) | **0.0032** | 0.0038 | 0.0047 | **0.0047** | 0.07310 |
| $L = 16$ | 0.0036 | 0.0031 | **0.0042** | 0.0048 | 0.06867 |
| $L = 24$ | 0.0038 | 0.0033 | 0.0053 | 0.0050 | 0.05378 |
| $L = 32$ | **0.0032** | 0.0031 | 0.0051 | 0.0049 | 0.04856 |
| $L = 40$ | 0.0036 | **0.0028** | **0.0042** | 0.0051 | **0.03724** |

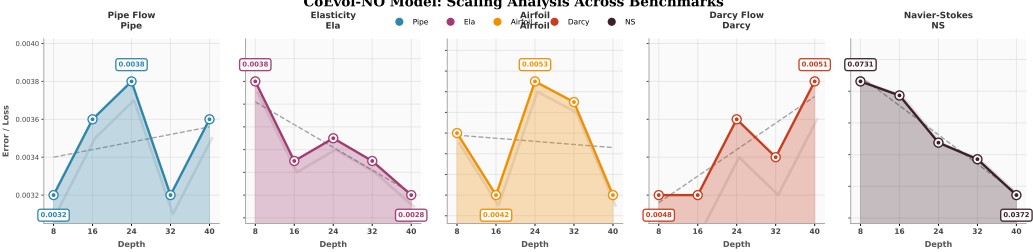

*Figure 4.* Scaling Analysis of Model Depth. Performance of CoEvol-NO as a function of the number of layers $L$. Results demonstrate significant scaling gains

**Observation 1: Significant Gain on Dynamic Tasks.** The most remarkable scaling behavior is observed in the **Navier-Stokes (NS)** dataset. As the depth increases from 8 to 40 layers, the relative error decreases monotonically from 0.07310 to 0.03724, representing a nearly **48% error reduction**. This indicates that our Predictor-Corrector framework effectively functions as an unrolled iterative solver (Jiang et al., 2025), where deeper architectures allow for more precise refinement of complex, time-dependent physical fields.

**Observation 2: Stability and Saturation on Steady Tasks.** For steady-state problems such as **Darcy** and **Pipe** flow, the performance remains relatively flat across different depths. This suggests that for simpler physical manifolds, CoEvol-NO reaches optimal convergence with fewer layers, demonstrating high parameter efficiency. The lack of performance degradation even at 40 layers further validates the stability of the error-driven update mechanism, which prevents the vanishing or exploding gradient issues often encountered in very deep operator transformers.

**Observation 3: Robustness to Noise.** The slight fluctuations in datasets like **Airfoil** and **Pipe** suggest that while depth generally helps, the model is also robust to hyperparameter variations within a reasonable range, consistently maintaining state-of-the-art performance levels regardless of the specific depth chosen beyond the baseline.

### H.2. Scaling with Latent Dimension $M$

We further investigate the impact of the latent dimension $M$, which governs the capacity of the persistent state $\mathbf{S}$ to encode physical dynamics. We evaluate $M \in \{32, 64, 128, 256, 512\}$ while keeping the depth fixed at $L = 8$. The results are summarized in Table 15.

*Table 15.* **Scaling Analysis of Latent Dimension $M$.** Relative $L_2$ error across benchmarks with varying latent sizes. The results indicate a trade-off between information capacity and optimization difficulty, with the optimal range typically falling between 128 and 256.

| Latent Dim ($M$) | Pipe | Elasticity | Airfoil | Darcy | Navier-Stokes |
|---|---|---|---|---|---|
| $M = 32$ | 0.0037 | 0.0039 | 0.0055 | 0.0075 | 0.0995 |
| $M = 64$ | 0.0036 | 0.0040 | 0.0050 | 0.0060 | 0.0905 |
| $M = 128$ (Default) | 0.0032 | 0.0038 | **0.0047** | **0.0047** | **0.0731** |
| $M = 256$ | **0.0030** | **0.0037** | 0.0051 | 0.0049 | 0.0785 |
| $M = 512$ | 0.0036 | 0.0040 | 0.0055 | **0.0047** | 0.1324 |

**Analysis of the Information Bottleneck.**

- **Rank Deficiency at Low Dimension ($M \leq 64$):** For all datasets, smaller latent dimensions result in higher errors. This can be theoretically explained by the rank constraints of the latent representation. Since the model's hidden channel dimension is fixed at $C = 128$, the algebraic rank of the latent state matrix $\mathbf{S} \in \mathbb{R}^{M \times C}$ is upper-bounded by $\min(M, C)$.

  Consequently, when $M < 128$ (e.g., $M = 32$ or $64$), the maximum possible rank is strictly limited by $M$. This explicitly restricts the model's capacity to span the feature space, preventing it from fully utilizing the channel dimensions to represent complex physical modes. This "low-rank truncation" leads to the observed underfitting.

- **The "Sweet Spot" ($M \in [128, 256]$):** Performance generally peaks around $M = 128$ or $M = 256$. This justifies our default hyperparameter choice of $M = 128$, which strikes an optimal balance between expressivity and computational efficiency.

- **Degradation at High Rank ($M = 512$):** Contrary to the intuition that "larger is better," increasing $M$ to 512 leads to performance saturation or even degradation, most notably in the **Navier-Stokes** task (Error increases from 0.0731 to 0.1324). This phenomenon suggests that an excessively large latent space may lead to overfitting or optimization difficulties, as the model attempts to learn a manifold that is higher-dimensional than the underlying physics requires. This aligns with the principle of parsimony in scientific modeling, where a compact representation often generalizes better.

*Table 16.* **Statistical Analysis over Multiple Runs.** We report the Mean $\pm$ Standard Deviation. The extremely low variance across all tasks confirms the training stability of the proposed Predictor-Corrector framework.

| BENCHMARK | RELATIVE $L_2$ ERROR (MEAN $\pm$ STD) |
|---|---|
| ELASTICITY | $0.0038 \pm 0.0001$ |
| AIRFOIL | $0.0047 \pm 0.0001$ |
| PIPE | $0.0032 \pm 0.0003$ |
| DARCY | $0.0047 \pm 0.0002$ |
| NAVIER-STOKES | $0.0731 \pm 0.0082$ |

*Figure 5.* **Evolutionary Dynamics of Latent States. Coords-Evol** shows disordered stagnation without physical inertia. **State-Evol** exhibits a rigid trajectory, implying limited convergence efficiency. In contrast, **CoEvol-NO** maintains a structured *"expansion-then-contraction"* manifold, demonstrating that our predictor-corrector mechanism facilitates a stable and robust physical evolution.

## I. Statistical Significance and Stability

To verify that the performance improvements of CoEvol-NO are robust and not artifacts of random initialization, we conducted repeated experiments with different random seeds. Table 16 reports the Mean and Standard Deviation (Std) of the relative $L_2$ error across the five standard PDE benchmarks.

**Analysis.** The results exhibit remarkable stability across all geometries. For static problems such as Elasticity and Airfoil, the standard deviation is on the order of $10^{-4}$, indicating that the model converges to a consistent optimum almost independently of the initial seed. Even for the challenging, time-dependent Navier-Stokes task, the variance remains small relative to the mean (0.0022 vs. 0.0731), confirming that the error-driven correction mechanism effectively stabilizes the learning trajectory against random perturbations.

## J. In-depth Visualization of Latent Dynamics

To intuitively understand the internal mechanism of CoEvol-NO, we visualize the layer-wise evolution of the latent state **S** using Principal Component Analysis (PCA). Figure 5 presents the projections of **S** across different layers (color-coded from Blue/L0 to Red/L8) for three distinct paradigms.

### J.1. Comparative Analysis of Architectural Paradigms

The PCA visualization reveals distinct evolutionary characteristics across the three paradigms, strictly corresponding to their architectural definitions.

For the **Coords-Evol** paradigm (Row 1), the latent states across different layers exhibit a high degree of overlap without a discernible evolutionary trajectory. This phenomenon stems from its design where the latent state **S** is typically initialized as

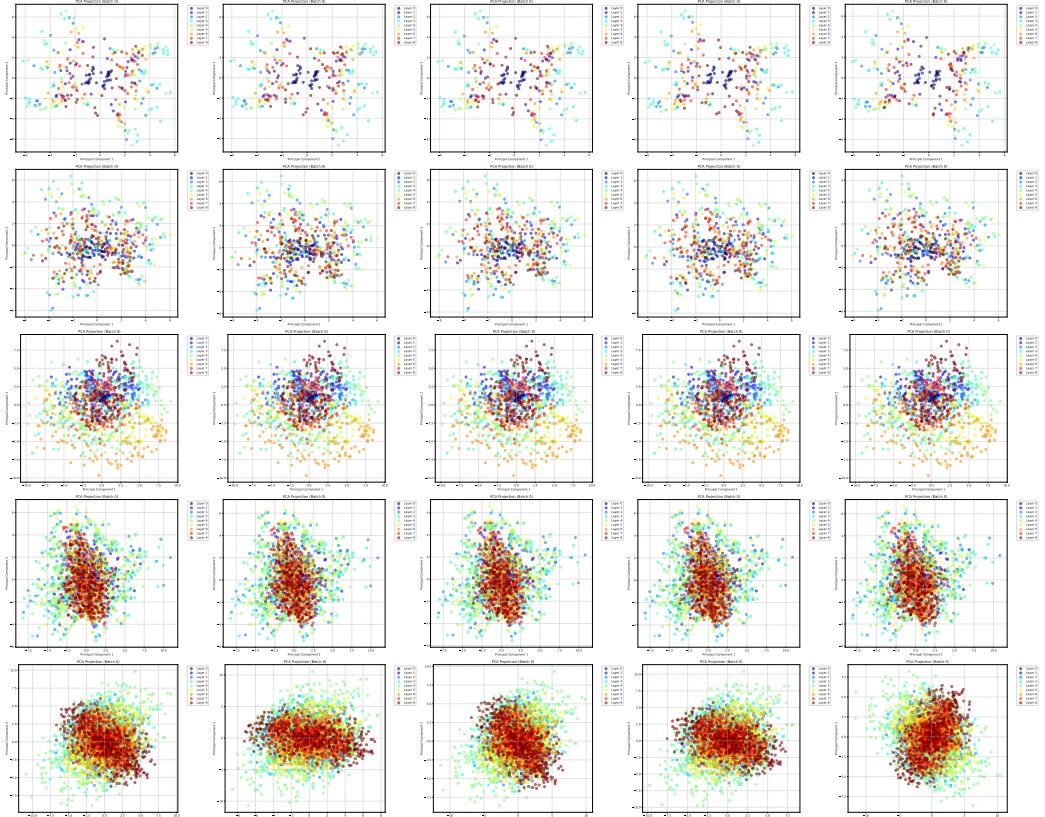

*Figure 6.* **Latent Evolutionary Dynamics under Varying Capacity** $M$**.** At the optimal dimension ($M = 128$), the state follows a disciplined *"expansion-then-contraction"* trajectory with high consistency across samples. In contrast, excessive capacity ($M = 512$) leads to fragmented and inconsistent evolutionary paths across different data batches. This visual divergence indicates that an overly broad latent space fails to converge to a unified invariant operator, aligning with the performance degradation observed in our scaling experiments.

fixed learnable parameters at each layer to compress the mesh information $\mathbf{X}$. Consequently, the state does not accumulate physical inertia from previous layers, appearing effectively static or "reset" at each step. This lack of temporal progression explains the model's inability to capture long-term dynamics in unsteady tasks.

In contrast, both the **State-Evol** (Row 2) and our CoEvol-NO (Row 3) paradigms demonstrate a clear, structured evolutionary pattern characterized by an **"Expansion-then-Contraction"** trajectory—expanding outward from the center and then converging back. This shared pattern indicates that both paradigms successfully model a continuous optimization process within the latent space.

However, a critical difference lies in the convergence behavior and consistency. The states in **State-Evol** remain highly tightly clustered within each layer. We hypothesize that this excessive stiffness in the trajectory may indicate a slower convergence rate, as the state evolution appears constrained and lacks the flexibility to traverse the manifold efficiently. On the other hand, CoEvol-NO maintains the regular "Expansion-then-Contraction" law while exhibiting remarkable consistency across different data samples. Since different input samples belong to the same physical task (governed by the same PDEs), their latent evolutions should theoretically follow a similar topological path. The visual alignment of different samples in CoEvol-NO confirms that it has successfully learned this intrinsic invariant operator, achieving a robust evolution that is both regularized and adaptive.

### J.2. Latent Evolution under Excessive Capacity ($M = 512$)

To investigate the performance degradation observed when the latent dimension is significantly increased, we visualize the evolution of $\mathbf{S}$ for $M = 512$. The PCA projections reveal a critical shift in the behavior of the latent manifold compared to lower-dimensional configurations.

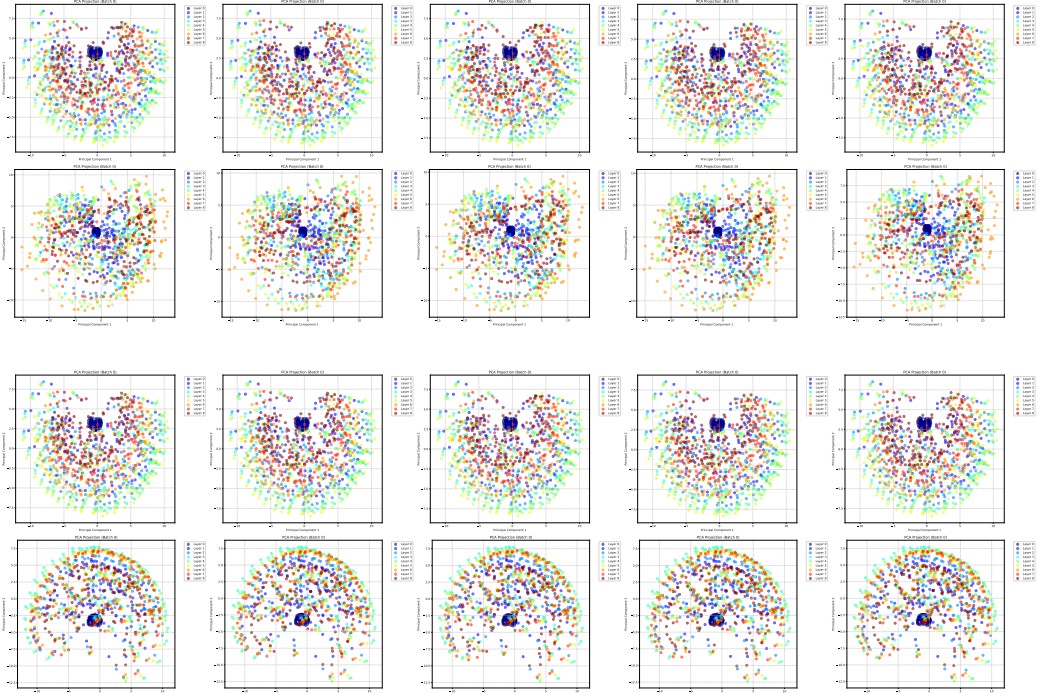

*Figure 7.* Varying Objectives and Update Mechanisms

A primary observation is the **lack of convergence to a unified intrinsic law** across different data samples. In the $M = 512$ case, the evolutionary trajectories for different batches exhibit significant discrepancies, appearing fragmented and inconsistent. This visual divergence suggests that the model has failed to capture the robust, shared physical manifold underlying the PDE task. Instead of learning a consistent operator that is invariant to specific input instances, the latent state in the $M = 512$ regime drifts along irregular paths. This lack of structural consistency across samples directly correlates with the sharp increase in the relative $L_2$ error (e.g., $0.1324$ for Navier-Stokes), indicating that the model is no longer effectively refining the physical state.

Furthermore, these results demonstrate that an **overly broad latent space does not necessarily lead to better outcomes**. While increasing $M$ initially provides the necessary capacity to overcome rank deficiency, an excessively large latent dimension (where $M \gg C$) creates a search space that is too vast for the Predictor-Corrector mechanism to regulate effectively. The absence of a compact "Expansion-then-Contraction" regularity in the $M = 512$ visualization confirms that when the latent capacity is too expansive, the model struggles to identify the core physical manifold, leading to optimization difficulties and a subsequent collapse in generalization performance.

### J.3. Visualizing Latent Dynamics under Varying Objectives and Update Mechanisms

To further evaluate the robustness of our framework, we visualize the latent state evolution under different objective functions—**Dot-Product Loss** vs. **Squared Loss**—and different update mechanisms—**Exact Gradient** vs. **First-order Approximation**. Across all these settings, the latent states consistently exhibit the distinct **"Expansion-then-Contraction"** trajectory, which we identified as the signature of a healthy and effective Co-Evolution process.

As shown in the visualizations, the latent states transition smoothly from the initialization (blue) to the final representation (red) along a well-defined manifold. This evolutionary regularity is maintained regardless of whether the update is driven by different loss priors or approximation strategies, and stands in stark contrast to the chaotic scattering observed in the Coords-Evol paradigm or the rigid clustering of the State-Evol paradigm.

These results provide compelling visual evidence that the **Predictor-Corrector framework** itself provides a robust regularization mechanism. It ensures that the state evolution remains stable, directional, and physically meaningful across various configurations. The high degree of regularity and the persistence of the "Expansion-then-Contraction" law corroborate the model's insensitivity to specific implementation details, highlighting the intrinsic stability and structural

superiority of the proposed co-evolutionary architecture.

### J.4. Baselines

We benchmark CoEvol-NO against a wide range of state-of-the-art neural operators:

**Baselines and Architectural Paradigms.** To rigorously evaluate the proposed Co-Evolution framework, we benchmark against representative methods from two distinct linear-complexity paradigms:

**1. Coordinate-Evolution (Coords-Evol).** These methods perform attention approximations directly on the discretized mesh representation but typically reset their features at each layer (stateless).

- **Transolver** (Wu et al., 2024) employs slice-based attention to capture local geometric correlations.

- **Set Transformer** (Lee et al., 2019) and **Nyströmformer** (Xiong et al., 2021) utilize inducing points and Nyström landmarks, respectively, to provide low-rank approximations of the full spatial interaction.

- **Fast Transformers** (Vyas et al., 2020) (with Clustered Attention) dynamically group queries using Locality Sensitive Hashing (LSH) to approximate spatial dependencies.

- **Lamo** (Tiwari et al., 2025) adopts a latent modeling approach for physical fields. In our experiments, we utilize the structured version of Lamo instead of the regular version to ensure a fair comparison, as the structured version is specifically designed to handle the geometric constraints present in our benchmarks.

**2. State-Evolution (State-Evol).** These methods compress the input into a compact latent space for deep processing, decoupling the evolution from the mesh (geometry-blind).

- **LNO** (Latent Neural Operator) (Wang & Wang, 2024) and **Perceiver** (Jaegle et al., 2021) project inputs onto a set of learnable latent tokens, evolving them in isolation before decoding. We specifically distinguish between the standard **Perceiver**, which evolves tokens in isolation via internal *self-attention*, and **Perceiver w/ Cross**, which utilizes *cross-attention* in each latent block to repeatedly query the initial mesh $X$ for information replenishment.

- **TokenLearner** (Ryoo et al., 2021) adaptively aggregates spatial features into a small set of tokens, effectively acting as a dynamic bottleneck.

As we will demonstrate, Coords-Evol methods struggle with long-term dynamics due to dynamic memory deficiency, while State-Evol methods fail to preserve fine-grained boundary conditions due to spatial decoupling. Our CoEvol-NO bridges this gap. For fair comparison, we utilize the official implementations and hyperparameters provided in their respective papers or the `NeuroFluid`/`NeuralOperator` libraries.

### J.5. Implementation Details and Hyperparameters

CoEvol-NO is implemented in PyTorch. All experiments are conducted on a NVIDIA H200 GPU. The training strategy largely follows the settings in Transolver (Wu et al., 2024) to ensure a fair comparison. We use the AdamW optimizer with One-CycleLR scheduler.

Architecture Configuration: The key hyperparameters for CoEvol-NO include:

- $L$: Number of layers (Predictor-Corrector blocks).

- $C$: Hidden channel dimension.

- $M$: Dimension of the persistent latent state $\mathbf{S}$.

- $\beta$: Momentum decay factor for the state evolution (typically set to 0.9).

Table 17 details the specific configurations for each benchmark.

*Table 17.* Hyperparameter settings for CoEvol-NO across all benchmarks. Note that we achieve SOTA performance with comparable or fewer parameters than baselines.

| BENCHMARK | TRAINING CONFIGURATION | | | | | MODEL CONFIGURATION | | |
|---|---|---|---|---|---|---|---|---|
| | EPOCHS | BATCH | LR | OPTIMIZER | SCHEDULER | $L$ (LAYERS) | $C$ (DIM) | $M$ (LATENT) |
| ELASTICITY | 500 | 1 | $10^{-3}$ | AdamW | Cosine | 8 | 128 | 128 |
| AIRFOIL | 500 | 4 | $10^{-3}$ | AdamW | OneCycleLR | 8 | 128 | 128 |
| PIPE | 500 | 8 | $10^{-3}$ | AdamW | OneCycleLR | 8 | 128 | 128 |
| NAVIER-STOKES | 500 | 2 | $10^{-3}$ | AdamW | OneCycleLR | 8 | 128 | 128 |
| DARCY | 500 | 4 | $10^{-3}$ | AdamW | OneCycleLR | 8 | 128 | 128 |
| SHAPE-NET CAR | 200 | 1 | $10^{-3}$ | Adam | OneCycleLR | 8 | 128 | 128 |
| AIRFRANS | 398 | 1 | $10^{-3}$ | Adam | OneCycleLR | 8 | 128 | 128 |

**Complexity Analysis vs. Baselines.**   CoEvol-NO maintains a strict linear complexity of $\mathcal{O}(NMC)$ per layer. Compared to Transolver, which reconstructs slice tokens from scratch at every layer ($\mathcal{O}(NMC)$), CoEvol-NO evolves a persistent state. The additional cost comes from the exact gradient calculation (Jacobian vector product), but as shown in the main text, this leads to faster convergence and better stability, effectively balancing the computational budget.

## K. Experimental Details

In this section, we provide a comprehensive description of the experimental setup, including dataset configurations, baseline implementations, evaluation metrics, and the specific hyperparameter settings used for CoEvol-NO.

### K.1. Benchmarks

We evaluate CoEvol-NO across eight challenging benchmarks that cover a wide range of physical phenomena, geometric complexities, and dimensions. These benchmarks are categorized into three types: Solid Mechanics (Elasticity), Fluid Dynamics (Airfoil, Pipe, Navier-Stokes, Shape-Net Car, AirfRANS), and Flow in Porous Media (Darcy). A summary of the dataset statistics is provided in Table 18.

- Elasticity (Solid, 2D): Estimates the internal stress of elastic materials with varying structural cutouts. The geometry is discretized into unstructured meshes with approximately 972 nodes.

- Airfoil (Fluid, 2D): Estimates the Mach number field around varying airfoil shapes in a transonic flow regime.

- Pipe (Fluid, 2D): Predicts the horizontal velocity field of fluids inside pipes with varying centerlines.

- Navier-Stokes (Fluid, 2D+Time): Models the incompressible viscous flow on a torus grid ($64 \times 64$). The task is to predict the fluid velocity in the next 10 timesteps given the past 10 timesteps. This is a crucial benchmark for evaluating the Co-Evolutionary mechanism's ability to capture non-stationary dynamics.

- Darcy (Flow, 2D): Models the steady-state pressure field of fluid flow through porous media. We use the standard $85 \times 85$ resolution setting.

- Shape-Net Car (Fluid, 3D): A large-scale industrial benchmark involving 3D flow around car shapes. The goal is to estimate surface pressure and the surrounding velocity field. The mesh contains approx. 32,186 nodes.

- AirfRANS (Fluid, 2D): A high-fidelity aerodynamic dataset based on RANS simulations. It involves varying Reynolds numbers and angles of attack, requiring the model to generalize across different physical conditions. For OOD evaluation, we use extrapolation settings (Wu et al., 2024): for Reynolds number, the model is trained on $Re \in [3, 5] \times 10^6$ and tested on $Re \in [2, 3] \cup [5, 6] \times 10^6$; for Angle of Attack, trained on $AoA \in [-2.5°, 12.5°]$ and tested on $AoA \in [-5°, -2.5°] \cup [12.5°, 15°]$.

*Table 18.* Summary of experimental benchmarks. #Mesh indicates the average number of discretized mesh points.

| GEOMETRY | BENCHMARKS | DIM | MESH | INPUT | OUTPUT |
|---|---|---|---|---|---|
| POINT CLOUD | ELASTICITY | 2D | 972 | STRUCTURE | STRESS |
| STRUCTURED MESH | AIRFOIL | 2D | 11,271 | STRUCTURE | MACH NUMBER |
| | PIPE | 2D | 16,641 | STRUCTURE | VELOCITY |
| REGULAR GRID | NAVIER-STOKES | 2D+Time | 4,096 | PAST VELOCITY | FUTURE VELOCITY |
| | DARCY | 2D | 7,225 | STRUCTURE | PRESSURE |
| UNSTRUCTURED MESH | SHAPE-NET CAR | 3D | 32,186 | STRUCTURE | VELOCITY & PRESSURE |
| | AIRFRANS | 2D | 32,000 | STRUCTURE | VELOCITY & PRESSURE |

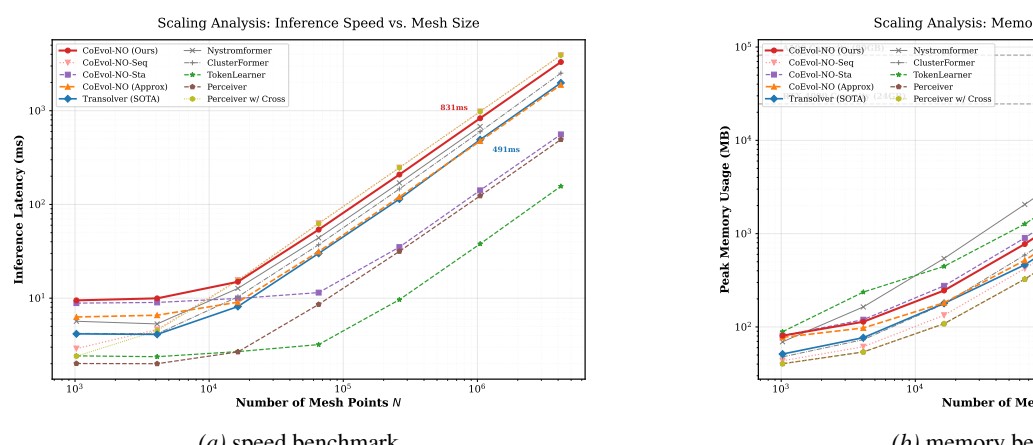

*(a)* speed benchmark                    *(b)* memory benchmark

*Figure 8.* Efficiency and Scalability Analysis on Large-Scale Meshes

## K.2. Evaluation Metrics

Following standard protocols (Wu et al., 2024), we employ the following metrics:

1. Relative L2 Error (Rel. L2): Used for all physics field predictions.

$$\text{Rel. L2} = \frac{\|\hat{\mathbf{u}} - \mathbf{u}\|_2}{\|\mathbf{u}\|_2}, \tag{43}$$

where $\hat{\mathbf{u}}$ and $\mathbf{u}$ denote the predicted and ground-truth fields, respectively.

2. Force Coefficients Error: For industrial design tasks (Shape-Net Car and AirfRANS), we evaluate the Drag Coefficient ($C_D$) and Lift Coefficient ($C_L$). These are integral quantities derived from the predicted surface pressure and shear stress fields. We report the relative L2 error of these coefficients.

3. Spearman's Rank Correlation ($\rho$): To assess the model's utility in design optimization, we calculate the Spearman's rank correlation between the ground-truth and predicted force coefficients. A value closer to 1 indicates better ranking capability.

## L. Efficiency Analysis

To rigorous evaluate the computational feasibility of CoEvol-NO for industrial-scale simulations, we conducted a comprehensive benchmarking of inference latency and memory consumption. All experiments were performed on a single **NVIDIA H200 GPU**, with the number of mesh points $N$ scaling from $10^3$ to $4 \times 10^6$. We compare our method against state-of-the-art linear attention mechanisms, including Transolver and Nyströmformer.

**Inference Latency (Strict $O(N)$).** As shown in Figure 8(a), all CoEvol-NO variants exhibit strict linear scaling parallel to Transolver. The `Dual` variant introduces a consistent $\sim 1.7\times$ constant overhead compared to Transolver (e.g., 831ms vs. 490ms at $10^6$ points), representing the necessary cost for the Exact Gradient mechanism. Notably, the `Approx` variant (first-order) matches Transolver's speed, confirming the efficiency of our backbone architecture.

**Memory Feasibility.** Figure 8(b) confirms that memory consumption scales linearly. At the extreme scale of **4 million points**, `CoEvol-NO-Dual` consumes ∼45GB VRAM, which comfortably fits within the capacity of standard A100 (80GB) or H200 GPUs. Furthermore, the `Seq` variant offers a lightweight alternative ( 24GB at 4M points), demonstrating superior memory efficiency compared to baselines.

## M. Runtime and Memory Benchmarks

We report per-epoch wall-clock time (seconds) and peak GPU memory (MB) on a single NVIDIA H100 with $M = 128$ latent tokens.

*Table 19.* Runtime and memory comparison. "Analytical" denotes the closed-form exact gradient implementation that avoids backpropagation graph construction.

| | AIRFOIL | | PIPE | | DARCY | |
|---|---|---|---|---|---|---|
| CONFIGURATION | TIME | MEM | TIME | MEM | TIME | MEM |
| CoEvol-NO (exact $S$, appro $X$) | 37.0 | 9,792 | 43.4 | 26,456 | 74.0 | 6,864 |
| CoEvol-NO (analytical, appro $X$) | 26.8 | 8,240 | 30.4 | 21,974 | 58.8 | 5,574 |
| CoEvol-NO (appro $S$, appro $X$) | 19.4 | 5,286 | 21.4 | 13,328 | 46.3 | 3,626 |
| CoEvol-NO (appro $S$, exact $X$) | 38.6 | 11,686 | 47.6 | 32,226 | 85.0 | 8,432 |
| TRANSOLVER | 14.2 | 5,062 | 17.7 | 12,436 | 37.6 | 3,204 |

Since the Cross-Attention predictor's exact gradient admits a closed-form expression, an analytical implementation avoids backpropagation graph construction, reducing the overhead to approximately $1.8\times$ Transolver's runtime while maintaining the accuracy benefits of exact gradients. The default backprop-based implementation has approximately $2.6\times$ overhead.

