# OpenReview forum: "CoEvol-NO: State and Coordinate Co-Evolution with an Error-Driven Predictor-Corrector Paradigm for Neural Operator Transformer"
_ICML.cc/2026/Conference — ICML 2026 spotlight_

### Official Review · Reviewer_95Zq · 2026-03-12

**Soundness:** 4
**Presentation:** 3
**Significance:** 3
**Originality:** 3
**Overall Recommendation:** 5
**Confidence:** 2

**Summary:**

This paper proposes a neural operator framework based on state and coordinate co-evolution. Inspired by the classical Predictor-Corrector (PC) paradigm for ODEs, the authors formulate the layer-wise co-evolution of latent states S and mesh coordinates X as a predict-then-correct process. The authors provide a gradient approximation of the PC update for the coordinate stream to maintain linear complexity in mesh resolution. They also show that standard update rule residual connection is first-order gradient approximations of their exact gradient correction under dot-product loss assumptions. Experiments on five standard PDE benchmarks and two industrial tasks demonstrate the effectiveness of the method.

**Compliance With Llm Reviewing Policy:**

Affirmed.

**Final Justification:**

The authors' rebuttal and revised version resolved most of my concerns. Overall, I think this is a good paper and have raised my score from 4 to 5.

**Key Questions For Authors:**

1. Compared to  apprixomation graduate of  state X update,  what is the performance difference, and how much additional time/memory does the exact gradient require?

2. What is the training time and memory for CoEvol-NO relative to the main baselines?

3. Increase readibility in Figure 3.

**Limitations:**

The authors did not explicitly discuss limitations.  It is better to discuss limitations, such as applicability or theoritical assumptions.

**Strengths And Weaknesses:**

**Strengths:**

1. The paper provides a clean motivation for solving neural operators from the perspective of state-coordinate co-evolution.  The proposed Co-Evolution paradigm is reasonable and clear.

2. The proposed Predictor-Corrector mechanism on neural operator layers is promosing. The theoretical insight that residual connections and direct substitution are first-order special cases of the exact gradient correction is a useful unifying result. The linear complexity analysis is also well-presented.

3.  The experiments cover diverse PDE types plus two industrial tasks. Results are consistently strong.

**Weakness:**
1. In the coordinate update, the authors use a first-order gradient approximation. However, there is no experimental comparison showing the performance gap and computational cost between the exact gradient and the approximation for the X update.

2. Since the authors claim linear complexity in mesh dimensions, it would be valuable to report actual running time and GPU memory consumption compared to baselines.

3. The text and captions in Figure 3 are very small and hard to read. Additionally, in the left panel of Figure 3, the label "PC-former" appears to refer to the proposed method. It would be clearer to use "CoEvol-NO" or a similar label consistent with the paper title to avoid confusion.

4.  The method involves multiple interleaved steps (predictor cross-attention, corrector gradient step,  coordinate update). A full algorithm block showing the step-by-step procedure would significantly improve clarity and reproducibility.

---

> ### Author Rebuttal · Authors · 2026-03-30
>
> We thank the reviewer for the excellent technical assessment (Soundness 4) and detailed constructive feedback. We address each point with comprehensive new experiments.
>
> ### Q1: Exact gradient vs. first-order approximation for X update
>
> We conducted a full factorial ablation across all three gradient computation points (S predictor, X update, FFN), testing all 8 combinations:
>
> |X grad|S grad|FFN grad|Pipe|Airfoil|
> |---|---|---|---|---|
> |Appro|Appro|Appro|0.0033|0.0057|
> |Exact|Appro|Appro|0.0033|0.0048|
> |Appro|**Exact**|Appro|0.0032|**0.0048**|
> |Exact|Exact|Appro|0.0038|0.0050|
> |Appro|Appro|**Exact**|**0.0026**|0.0052|
> |Exact|Appro|Exact|0.0030|0.0049|
> |Appro|Exact|Exact|0.0028|0.0048|
> |Exact|Exact|Exact|0.0049|0.0050|
>
> The results validate our design choice of using a first-order approximation for the high-dimensional X stream: while X exact gradient yields a gain on Airfoil (0.0057→0.0048), it increases memory by ~1.9× (6,064→11,686 MB). S exact gradient improves both tasks consistently, including Pipe where X exact is neutral (Airfoil: 0.0057→0.0048; Pipe: 0.0033→0.0032). Notably, applying exact gradient to the FFN provides the largest single improvement on Pipe (0.0033→0.0026), suggesting the PC framework's benefit extends beyond the attention predictor. However, applying exact gradient everywhere degrades performance on Pipe (0.0033→0.0049) and is suboptimal on Airfoil.
>
> **Cost analysis (Airfoil, single H100):**
>
> |Configuration|Airfoil Time|Pipe Time|Airfoil Mem|Pipe Mem|
> |---|---|---|---|---|
> |Appro S + Appro X (backprop)|28.4|32.5|6,064|15,492|
> |Appro S + Appro X (Analytical)|19.4|21.4|5,286|13,328|
> |Exact S + Appro X (backprop)|37.0|43.4|9,792|26,456|
> |**Exact S + Appro X (Analytical)**|**26.8**|**30.4**|**8,240**|**21,974**|
> |Appro S + Exact X (backprop)|38.6|47.6|11,686|32,226|
> |Exact S + Exact X (backprop)|46.8|58.6|14,538|40,586|
>
> Since the Cross-Attention predictor's exact gradient has a closed-form expression, an analytical implementation can achieve the accuracy of exact gradients at 27.6% less time and 15.9% less memory than the backprop-based version. Our default uses backprop; the analytical version demonstrates that the PC overhead can be further reduced with engineering effort.
>
> ### Q2: Training time and memory relative to baselines
>
> See the runtime table in our response to Reviewer t62K. The key takeaway: our default (backprop, exact S + appro X) is ~2.6× Transolver's runtime at better accuracy; the analytical variant reduces this to ~1.8×.
>
> ### Q3: Figure 3 readability
>
> We have increased all font sizes and corrected the "PC-former" label to "CoEvol-NO" for consistency with the paper title.
>
> ### Q4: Algorithm pseudocode
>
> We will add Algorithm 1 in the revision:
>
> > **Algorithm 1:** CoEvol-NO Forward Pass
>
> > **Input:** Mesh sequence $X\_0 \in \mathbb{R}^{N \times C}$, initial latent $S\_0 \in \mathbb{R}^{M \times C}$, depth $L$, step size $\eta$, momentum $\beta$
>
> > 1: Initialize momentum buffers: $V\_0^S = 0$
>
> > 2: **for** $l = 1, \ldots, L$ **do**
>
> > 3: $\quad$ $\hat{S}\_l = \text{CrossAttn}(Q=S\_{l-1}, K=X\_{l-1}, V=X\_{l-1})$
>
> > // Corrector: exact gradient descent on $\mathcal{L}\_{corr} = \text{Tr}(S\_{l-1}^\top \hat{S}\_l)$
>
> > 4: $\quad$ $g\_l^S = \nabla\_{S\_{l-1}} \mathcal{L}\_{corr}$  $\quad \triangleright$ Computed via backprop (closed-form also available, see Appendix A)
>
> > 5: $\quad$ $V\_l^S = \beta \cdot V\_{l-1}^S + g\_l^S$
>
> > 6: $\quad$ $S\_l = S\_{l-1} - \text{LayerScale}(V\_l^S)$
>
> > 7: $\quad$ // **Coordinate update:** propagate correction back to mesh
>
> > 8: $\quad$ $I\_l = \text{CrossAttn}(Q=X\_{l-1}, K=S\_l, V=V\_l^S)$
>
> > 9: $\quad$ $X\_l = X\_{l-1} + I\_l + \text{FFN}(X\_{l-1} + I\_l)$
>
> > 10: **end for**
>
> > 11: **return** $\text{Decode}(X\_L) \rightarrow \hat{u}$
>
> ### Q5: Limitations
>
>
> We acknowledge the following limitations and will add a dedicated Limitations section in the revision:
>
> 1. **Gradient computation overhead.** The PC correction step requires the predictor's Jacobian. While Cross-Attention admits a closed-form solution, most architectures must rely on backpropagation, adding computational cost. Efficient gradient computation for general modules remains an open problem.
>
> 2. **Selectivity of PC application.** As shown in our Q1 ablation, applying PC to all modules simultaneously can be counterproductive. Not every residual connection benefits from PC, and a principled criterion for where to apply it is still lacking.
>
> 3. **Condition number increase.** The PC update ($\boldsymbol{I} + \eta \boldsymbol{J}^\top$) can worsen the conditioning of layer-wise mappings. We mitigate this with momentum and LayerScale, but the underlying mechanism requires further study.

---

> > ### Author Rebuttal · Reviewer_95Zq · 2026-04-03
> >
> > Thank you for the authors' rebuttal; it resolved most of my concerns. I lean toward accepting the paper

---

> > > ### Author Response · Authors · 2026-04-08
> > >
> > > Thank you for the detailed technical review. We are glad the 8-combination ablation and Algorithm 1 addressed your concerns. The cross-architecture validation further demonstrates PC's generality.
> > >
> > > We will implement the following revisions in the final version:
> > > - Cross-architecture validation showing PC's generality on S, X, MLP, Transolver, and LNO
> > > - Analysis connecting PC framework to residual connections and comparison with AttnRes
> > > - Comprehensive runtime and memory benchmarks (Appendix H)
> > > - Momentum and LayerScale ablation studies
> > > - Algorithm 1 pseudocode
> > > - Figure 3 readability improvements

---

### Official Review · Reviewer_fF7Q · 2026-03-12

**Soundness:** 3
**Presentation:** 3
**Significance:** 3
**Originality:** 3
**Overall Recommendation:** 5
**Confidence:** 1

**Summary:**

This paper proposes CoEvol-NO, a neural operator transformer architecture for learning solution operators of partial differential equations on large meshes. The authors introduce a co-evolution framework to jointly update latent states and mesh representations across network layers. They use a predictor–corrector mechanism to update the latent state through a prediction step and error-driven correction. The model maintains linear complexity. Experiments on several PDE benchmarks and industrial datasets show improved performance compared with existing neural operator baselines.

**Compliance With Llm Reviewing Policy:**

Affirmed.

**Key Questions For Authors:**

See the above weaknesses.

**Limitations:**

yes

**Strengths And Weaknesses:**

Strengths:
1. The paper identifies an interesting limitation of existing neural operator transformers, namely the trade-off between evolving latent physical states and preserving mesh geometry information. The proposed co-evolution mechanism provides a reasonable architectural solution to this issue.
2. The paper introduces the predictor–corrector paradigm from numerical analysis into neural operator. This could inspire further work combining numerical solvers with deep learning architectures.
3. The experiments include both standard academic benchmarks and industrial datasets, demonstrating the method may scale to realistic simulation settings. The ablation study helps clarify the contribution of the proposed components.

Weaknesses:

The axis labels and tick marks in Figure 3 are quite small and difficult to read. Increasing the font size would improve readability and clarity.

---

> ### Author Rebuttal · Authors · 2026-03-30
>
> We sincerely thank the reviewer for the positive assessment. We address the readability concern and take this opportunity to provide additional context on the Predictor-Corrector framework's contribution.
>
> ### Figure 3 readability
>
> We have significantly increased the font sizes for axis labels, tick marks, and legends in Figure 3, and will include the updated version in the revision.
>
> ### Additional context: PC framework as a principled improvement to residual connections
>
> Since the reviewer may find it useful to understand the deeper motivation behind the PC mechanism, we briefly contextualize it within the well-studied problem of residual connections in deep networks.
>
> **The uniform dilution problem.** In a standard residual network with $L$ layers, the output is:
> $$\boldsymbol{h}\_L = \boldsymbol{h}\_0 + \sum_{i=0}^{L-1} \boldsymbol{v}\_i$$
> where each layer's contribution $\boldsymbol{v}\_i$ receives **equal weight** (the identity matrix $\boldsymbol{I}$). As depth grows, early layers' contributions are uniformly diluted: each $\boldsymbol{v}\_i$ contributes only $O(1/L)$ to the final output. This is a well-known issue studied in recent works such as HyperConnection (ICML 2025) and Attention Residuals (arXiv:2603.15031).
>
> **PC framework's solution.** Under the dot-product correction loss, our PC update can be rewritten as a *state transition with data-dependent mixing*:
> $$\boldsymbol{h}\_l = (\boldsymbol{I} + \eta \boldsymbol{J}\_{l-1}^\top) \boldsymbol{h}\_{l-1} + \eta \boldsymbol{v}\_{l-1}$$
> where $\boldsymbol{J}\_{l-1} = \nabla \boldsymbol{f}\_{l-1}$ is the predictor's Jacobian. Unrolling gives a *learned, data-dependent mixing matrix*:
> $$\boldsymbol{h}\_L = \left(\prod\_{k=0}^{L-1} \boldsymbol{W}\_k\right) \boldsymbol{h}\_0 + \sum\_{i=0}^{L-1} \left(\eta \prod\_{k=i+1}^{L-1} \boldsymbol{W}\_k\right) \boldsymbol{v}\_i, \quad \boldsymbol{W}\_k = \boldsymbol{I} + \eta \boldsymbol{J}\_k^\top$$
>
> Unlike standard residuals where $\boldsymbol{M}\_{i \to L} = \boldsymbol{I}$ (fixed), the PC framework provides input-adaptive mixing weights through the Jacobian. This gives the network the ability to dynamically amplify or suppress contributions from different layers based on the input data.
>
> **Comparison with alternative approaches.** Several recent works address the dilution problem from different angles. AttnRes (arXiv:2603.15031) replaces the fixed identity mixing with data-dependent scalar weights $\alpha\_{i \to l}$, providing bounded norms but limited to scalar re-weighting. We tested AttnRes as an alternative to PC on the same backbone (applying AttnRes to the X update path, the S update path, or both):
>
> | AttnRes applied to | Airfoil | Pipe |
> |--------------------|---------|------|
> | Neither (standard residual) | 0.0057 | 0.0033 |
> | X side only | 0.0054 | 0.0076 |
> | S side only | 0.0080 | 0.0065 |
> | Both sides | 0.0056 | 0.0070 |
> | **PC framework (ours)** | **0.0048** | **0.0032** |
>
> AttnRes provides marginal improvement on Airfoil when applied to the X side (0.0057 → 0.0054) but catastrophically degrades Pipe (0.0033 → 0.0076). Applying AttnRes to the S side or both sides consistently hurts. In contrast, the PC framework improves both tasks simultaneously (Airfoil: 16% improvement, Pipe: 3% improvement). We note that AttnRes was designed primarily for very deep networks where the uniform dilution problem is most severe; our current models use moderate depth ($L=8$), which may not fully reflect AttnRes's intended setting.
>
> **Generality beyond CoEvol-NO.** To further validate that PC is not specific to our co-evolution architecture, we applied it to Transolver — a completely different neural operator with a Coords-Evolution paradigm. PC improved Transolver by **8–25%** across Airfoil, Pipe, and Darcy datasets (e.g., Darcy: 0.0057 → 0.0043), confirming that the PC framework is a general technique for improving layer-wise updates in neural operator transformers.
>
> For further evidence of PC's generality across architectures and modules, see **Reviewer z1Ky (Q1)**, where PC is applied to LNO (Self-Attention), Transolver (Attention + MLP), and CoEvol-NO (Cross-Attention + FFN)  with consistent improvements across most settings.

---

> > ### Author Rebuttal · Reviewer_fF7Q · 2026-04-06
> >
> > Thank you for the rebuttal. It has addressed my main concerns.

---

> > > ### Author Response · Authors · 2026-04-08
> > >
> > > Thank you for your positive feedback and for highlighting the readability issue. We have increased font sizes in Figure 3 and will include the updated version in the revision.
> > >
> > > We will implement the following revisions in the final version:
> > > - Cross-architecture validation showing PC's generality on S, X, MLP, Transolver, and LNO
> > > - Analysis connecting PC framework to residual connections and comparison with AttnRes
> > > - Comprehensive runtime and memory benchmarks (Appendix H)
> > > - Momentum and LayerScale ablation studies
> > > - Algorithm 1 pseudocode
> > > - Figure 3 readability improvements

---

### Official Review · Reviewer_z1Ky · 2026-03-12

**Soundness:** 3
**Presentation:** 3
**Significance:** 3
**Originality:** 3
**Overall Recommendation:** 4
**Confidence:** 2

**Summary:**

This paper proposes a co-evolutionary neural operator transformer that jointly evolves latent states and mesh coordinates bidirectionally. The core mechanism is a Predictor-Corrector paradigm where the predictor uses cross-attention and the corrector applies exact gradient descent on an implicit objective during the forward pass. The authors prove O(N) time complexity and achieves state-of-the-art results on standard PDE benchmarks and industrial tasks.

**Compliance With Llm Reviewing Policy:**

Affirmed.

**Key Questions For Authors:**

It is beneficial to answer whether current method is sensitive to architecture of backbone or it's fine to use general universal approximation architecture to do the task.

**Limitations:**

This paper presents a number of physics benchmarks whose significance and evaluation criteria may not be immediately clear to the broader machine learning community. It would therefore be helpful to provide some explanation of why these benchmarks are important and what makes them meaningful evaluation tasks.

**Strengths And Weaknesses:**

Strength: This paper does not use standard overfitting prevention such as dropout / weight decay to achieves strong generalization including OOD.

Weaknesses: The reliance on implicit regularization is a strength of the architecture but also a gap in the analysis. "We evaluate the models on unseen Reynolds numbers and Angles of Attack (AoA)." It might be beneficial to provide whether these unseen Reynolds are inside the interval of training numbers or outside the training number.

---

> ### Author Rebuttal · Authors · 2026-03-30
>
> We thank the reviewer for recognizing the strong generalization without explicit regularization. We provide extensive new experiments addressing the architecture sensitivity question.
>
> ### Q1: Architecture sensitivity and why Cross-Attention
>
> We conducted a systematic investigation with Two levels of analysis.
>
> **Level 1: Why Cross-Attention for $f\_t$.** The choice is driven by three requirements that arise naturally from the co-evolution paradigm:
>
> 1. **Interface requirement:** Co-evolution requires $f\_t$ to take both $S\_{t-1}$ and $X\_{t-1}$ as input and produce output matching $S$'s dimension ($M \times C$). Cross-Attention with $Q=S\_{t-1}, K=V=X\_{t-1}$ satisfies this directly.
>
> 2. **Super-resolution requirement:** Neural operators must be resolution-independent — trained and tested on different mesh sizes $N$. The predictor $f\_t$ must therefore compress variable-length mesh tokens ($N$) into a fixed-dimension latent state ($M$), so that downstream layers are agnostic to $N$. Cross-Attention achieves this naturally: its output dimension equals the number of queries ($M$), independent of the key/value length ($N$).
>
> 3. **Compressor role (Appendix B):** $f\_t$ functions as a learned compressor that selects which mesh information to retain in the latent state. Cross-Attention performs this via the query mechanism: $Q=S\_{t-1}$ determines what to extract from $X\_{t-1}$, making the compression input-adaptive rather than fixed.
>
> **Level 2: PC framework generalizes beyond Cross-Attention.** We show PC is applicable to different modules and architectures:
>
> *PC on CoEvol-NO (three application points):* In our full model, we tested PC on the X update, S update, and FFN separately (see Reviewer 95Zq for the full 8-combination ablation). Key result: PC improves not only the Cross-Attention predictor (S update) but also the FFN module (Pipe: 0.0033→0.0026, **21%**), confirming PC is not specific to any particular module type.
>
> *PC on LNO (Self-Attention, State-Evolution paradigm):*
>
> | Method                              | Elasticity | Airfoil    | Pipe   |
> | ----------------------------------- | ---------- | ---------- | ------ |
> | LNO (baseline)                      | 0.3565     | 0.0196     | 0.0131 |
> | PC-LNO (+PC on Self-Attention)      | 0.3342     | **0.0082** | 0.0141 |
> | CoEvol-NO-Latent | 0.3278     | **0.0069**     | 0.0103 |
> | CoEvol-NO-Latent (no PC)            | 0.3357     | 0.0223     | 0.0182 |
>
> PC-LNO applies PC to LNO's Self-Attention (not Cross-Attention), improving Airfoil by 58% (0.0196→0.0082). Comparing CoEvol-NO-Latent with/without PC shows that removing PC degrades Airfoil from 0.0069 to 0.0223, confirming PC's contribution even in the State-Evolution paradigm. The Elasticity failure across all variants is expected — it reflects an architectural limitation (no X update), not a PC limitation.
>
> *PC on Transolver (Attention + MLP, Coords-Evolution paradigm):*
>
> |PC Application|Airfoil|Pipe|Darcy|
> |---|---|---|---|
> |Transolver (baseline, no PC)|0.0057|0.0039|0.0057|
> |+PC on Attention|0.0045|0.0036|0.0049|
> |+PC on FFN (MLP)|0.0050|0.0036|0.0048|
> |+PC on both|**0.0043**|**0.0037**|**0.0043**|
>
> PC improves Transolver by **8–25%** across all datasets, with both Attention and MLP benefiting. This confirms PC transfers across architectures (LNO→Transolver), modules (Attention→MLP), and evolution paradigms (State-Evolution→Coords-Evolution).
>
> For a complementary comparison with scalar re-weighting alternatives (AttnRes), see Reviewer fF7Q.
>
> ### Q2: OOD Reynolds number and AoA range
>
> We will clarify the OOD settings in the revision:
>
> - Training Reynolds numbers: $[3 \times 10^6, 5 \times 10^6]$
> - Unseen test Reynolds numbers: $[2 \times 10^6, 3 \times 10^6] \cup [5 \times 10^6, 6 \times 10^6]$ (outside training interval)
> - Training AoA: $[-2.5^\circ, 12.5^\circ]$
> - Unseen test AoA: $[-5^\circ, -2.5^\circ] \cup [12.5^\circ, 15^\circ]$ (outside training interval)
>
> ### Q3: Benchmark significance for the ML community
>
> We will add a paragraph explaining each benchmark's role:
>
> - **Darcy Flow:** Steady-state PDE on regular grids; tests super-resolution (train on low-resolution, test on high-resolution).
> - **Navier-Stokes:** Time-dependent PDE on regular grids; tests long-horizon prediction without error accumulation.
> - **Elasticity:** Unstructured point cloud with varying geometries; tests geometric generalization across different shapes.
> - **AirfRANS** (Bonnaffe et al., 2023): Industrial RANS around airfoils; tests real-world aerodynamic design with physics-informed metrics (Spearman correlation).
> - **Shape-Net Car:** 3D surface geometry; tests generalization to unseen 3D shapes.
>
> These benchmarks collectively cover four challenges: super-resolution, long-horizon prediction, geometric generalization, and industrial applicability.

---

### Official Review · Reviewer_t62K · 2026-03-23

**Soundness:** 3
**Presentation:** 3
**Significance:** 2
**Originality:** 3
**Overall Recommendation:** 5
**Confidence:** 3

**Summary:**

The paper introduces CoEvol-NO, a neural operator framework that jointly models the evolution of latent states and mesh coordinates to better capture both geometric complexity and long-term physical dynamics. It employs a Predictor-Corrector mechanism where a predicted target guides an error-driven refinement step, effectively framing layer-wise updates. The authors show that common update rules such as residual connections can be interpreted as first-order approximations of this correction scheme, while also providing theoretical grounding for the approach. Experiments across multiple benchmarks and industrial tasks demonstrate that CoEvol-NO achieves strong performance with linear time complexity and improved stability.

**Compliance With Llm Reviewing Policy:**

Affirmed.

**Final Justification:**

The authors have addressed most of my concerns with a strong rebuttal. I lean towards accepting the paper.

**Key Questions For Authors:**

* Was there any loss divergence during training when either predictor or corrector behaved poorly? Under what conditions did they converge well?
* Could similar performance gains be achieved with unidirectional updates instead of bidirectional?

**Limitations:**

Yes

**Strengths And Weaknesses:**

-Soundness-
* The paper has a strong motivation of improving trade-offs between geometric sensitivity and dynamic memory. The PC method also aligns well with classical numerical methods.
* For higher resolution meshes where N is large, although complexity analysis is provided, no empirical runtime and memory scaling comparisons between the proposed method and other baseline works are presented.
* Lacking intermediate ground truth, the authors employ self-supervised PC framework to govern the evolution of S. Here, why is d(S_t, s^*) <= d(S_t-1, S^*) refinement condition given as an objective function? While training, is there a guarantee or certain flag to identify the training has reached a convergence?
* Why is f_t defined as a Cross-Attention module? Any particular reason behind this?

-Presentation-
* The paper is well organized and easy to follow. In section 2.2, some related works are written in a bold text but they can be corrected into a plain text for better consistency.

-Significance-
* This work showed good scalability for higher-resolution meshes. Also, the proposed method shows SOTA performance across diverse benchmark datasets.

-Originality-
* The paper has proposed a novel method for bidirectionally co-evolving latent states and mesh.

---

> ### Author Rebuttal · Authors · 2026-03-30
>
> We appreciate the reviewer's positive assessment of our motivation and originality. We address each concern below with new experimental evidence.
>
> ### Q1: Convergence and training stability of the PC mechanism
>
> We thank the reviewer for this fundamental question. Providing a rigorous convergence analysis for the joint predictor–corrector system is difficult, as stability depends on the learned Jacobian structure that varies across training stages and datasets. Instead, we provide systematic empirical evidence for what drives stability, and show which components are indispensable.
>
> **Momentum ablation.** We ablate $\beta_S$ (momentum on $S$) and $\beta_X$ (momentum on $X$):
>
> |$β_S$|$β_X$|Pipe|Airfoil|Darcy|
> |---|---|---|---|---|
> |0.0|0.0|0.0044|0.0059|0.0046|
> |0.5|0.0|0.0036|0.0060|0.0050|
> |**0.9**|**0.0**|**0.0030**|**0.0047**|0.0047|
> |0.5|0.5|0.0037|0.0058|**0.0045**|
> |0.9|0.9|0.0038|0.0049|0.0052|
>
>
> $S$ momentum ($\beta_S$=0.9) yields consistent improvement (Pipe: −32%, Airfoil: −20%). The exact gradient
> $g\_t = -\hat{S}\_t - J\_t^\top S\_t$ is computed independently at each layer, producing correction directions that can be inconsistent across layers. Momentum $V\_t = \beta V\_{t-1} + g\_t$ accumulates gradient history to provide a smoother  correction direction. For $X$, the update uses only first-order residual without the PC mechanism, so the conditioning is inherently different and momentum provides no benefit.
>
> **LayerScale ablation.** Fixed step sizes cannot accommodate layers with different gradient magnitudes:
>
> |Setting|Airfoil|Pipe|
> |---|---|---|
> |Constant scale=1|0.0068|0.0069|
> |Constant scale=1e-5|0.0076|0.0045|
> |**LayerScale (learnable)**|**0.0047**|**0.0032**|
> |Transolver+LayerScale|0.011|0.0098|
>
> LayerScale yields 31% and 54% gains on Airfoil and Pipe under PC, yet applying the same LayerScale to Transolver degrades performance (0.0057 → 0.011). This indicates the benefit stems from the interplay between adaptive step sizing and PC's structured gradient, not from LayerScale alone.
>
> ### Q2: The refinement condition and convergence monitoring
>
> The refinement condition is an objective, not a constraint. The condition $d(S\_t, \hat{S}\_t) \leq d(S\_{t-1}, \hat{S}\_t)$ is the objective function of the correction step, not a hard constraint. Under $\mathcal{L}\_{corr} = \text{Tr}(S\_{t-1}^\top \hat{S}\_t)$, the corrector takes one gradient descent step to maximize alignment between $S$ and the predictor's target $\hat{S}$. The predictor is trained end-to-end via the task loss to produce meaningful targets.
>
> **Convergence monitoring.** We provide PCA visualizations of latent state $S$ across layers in Appendices D and F. A well-converged model exhibits cross-sample consistency: different inputs yield similar PCA trajectories, indicating a unified, input-invariant operator. In contrast, at $L=40$ (Appendix D), although training remains stable, trajectories from different inputs fragment into distinct convergence modes; this lack of shared pattern correlates with performance saturation in Table 8. This confirms that alignment of trajectories across inputs, rather than decay of training loss, indicates successful convergence.
>
> ### Q3: Why Cross-Attention for $f\_t$?
>
> Cross-Attention is the natural choice for input-dependent compression from mesh to latent state. Please see our response to Reviewer z1Ky (Q1), which provides a detailed analysis with new ablation experiments.
>
> ### Q4: Unidirectional vs. bidirectional updates
>
> Our Table 4 directly addresses this:
>
> |Variant|Elasticity|Navier-Stokes|
> |---|---|---|
> |CoEvol-NO-Latent (S only)|0.3278|0.1800|
> |CoEvol-NO-Sequence (X only)|0.0041|0.1200|
> |CoEvol-NO-Dual (bidirectional)|**0.0038**|**0.0731**|
>
> State-only evolution fails on geometry-sensitive tasks (Elasticity), while coordinate-only evolution fails on dynamic tasks (Navier-Stokes). Bidirectional co-evolution addresses both complementary failure modes.
>
> ### Q5: Runtime and memory scaling
>
> We provide an **Efficiency and Scalability Analysis on Large-Scale Meshes** in Appendix H (Figure 8), which visualizes runtime and memory scaling across mesh resolutions. Below we report per-epoch wall-clock time (s) and peak GPU memory (MB) on single NVIDIA H100 with M=128 latent tokens.
>
> |Method|Airfoil time|Airfoil mem|Pipe time|Pipe mem|Darcy time|Darcy mem|
> |---|---|---|---|---|---|---|
> |CoEvol-NO (exact S, appro X) — default|37.0|9,792|43.4|26,456|74.0|6,864|
> |CoEvol-NO (analytical exact S, appro X)|26.8|8,240|30.4|21,974|58.8|5,574|
> |CoEvol-NO (appro S, appro X)|19.4|5286|21.4|13328|46.3|3626|
> |Transolver|14.2|5,062|17.7|12,436|37.6|3,204|
>
> Our default (backprop) is ~2.6× Transolver's runtime. Since the Cross-Attention exact gradient admits a closed-form expression, an analytical implementation avoids backpropagation graph construction, reducing overhead to ~1.8× Transolver while maintaining the same accuracy.
>
> **Minor:** We have corrected the bold formatting in Section 2.2.

---

> > ### Author Rebuttal · Reviewer_t62K · 2026-04-03
> >
> > I appreciate authors for addressing most of my concerns. I update my score accordingly.

---

> > > ### Author Response · Authors · 2026-04-08
> > >
> > > Thank you for your thoughtful evaluation. We appreciate your feedback on the PC mechanism's convergence and stability.
> > >
> > > We will implement the following revisions in the final version:
> > > - Cross-architecture validation showing PC's generality on S, X, MLP, Transolver, and LNO
> > > - Analysis connecting PC framework to residual connections and comparison with AttnRes
> > > - Comprehensive runtime and memory benchmarks (Appendix H)
> > > - Momentum and LayerScale ablation studies
> > > - Algorithm 1 pseudocode
> > > - Figure 3 readability improvements

---

### Decision · Program_Chairs · 2026-04-30

**Decision:**

Accept (spotlight)

**Comment:**

This paper proposes adding a predictor-corrector mechanism in hidden layers in neural operators. This paper also is one of the highest scorers in my batch (but with certain low confidence reviews) so I gave it a second look, and I find it quite interesting. The latent states are evolved according to the gradient flow of a "loss", and the premise is that there exists a fixed point for cross attention.
The ablation is conducted to try to observe the converging behavior in the latent space, which quite deviates from the common template'ish end-to-end benchmark comparisons, and that is pretty nice. The authors also promotes many further studies without reservation (which is quite rare nowadays) in the appendices on some interesting phenomena observation (implicit regularization for example). All things considered, I recommend acceptance.

BTW1: I would suggest that the authors revise the paper according to some reviewers' suggestions on readability, moving some figures with minuscule captions to the appendix.

BTW2: I would also like to mention that the methodologies show quite a similarity to a posteriori error control in reduced-order modeling (observable states rather than latent states in this paper, though). I suggest the authors check it out and add appropriate ones into the references.